# Probabilistic program inference in network-based epidemiological simulations

**Niklas Smedemark-Margulies**[1☯]*, **Robin Walters**[1☯]*, **Heiko Zimmermann**[2], **Lucas Laird**[1,3], **Christian van der Loo**[3], **Neela Kaushik**[3], **Rajmonda Caceres**[3], **Jan-Willem van de Meent**[1,2]

**1** Khoury College of Computer Science, Northeastern University, Boston, Massachusetts, United States of America, **2** Informatics Institute, University of Amsterdam, Amsterdam, Netherlands, **3** MIT Lincoln Laboratory, Lexington, Massachusetts, United States of America

☯ These authors contributed equally to this work.
* smedemark-margulie.n@northeastern.edu (NSM); r.walters@northeastern.edu (RW)

**Data Availability Statement:** Code and data to reproduce all experiments is available at https://github.com/nik-sm/ProbProgEpiNet.jl. This

## Abstract

Accurate epidemiological models require parameter estimates that account for mobility patterns and social network structure. We demonstrate the effectiveness of probabilistic programming for parameter inference in these models. We consider an agent-based simulation that represents mobility networks as degree-corrected stochastic block models, whose parameters we estimate from cell phone co-location data. We then use probabilistic program inference methods to approximate the distribution over disease transmission parameters conditioned on reported cases and deaths. Our experiments demonstrate that the resulting models improve the quality of fit in multiple geographies relative to baselines that do not model network topology.

## Author summary

The ability to create computer simulations of epidemics is important to be able to predict where and when people will be become infected, identify factors which either contribute to or slow disease spread, and test various interventions without risking real lives. However, the conclusions of experiments performed using these simulations are only meaningful in the real world if we can be sure the simulation accurately models what is happening in the real world. We study methods for fitting parameters, such as infectiousness, to real world data so that the disease simulator correctly represents the actual disease. We achieve this using probabilistic programming methods which automatically adjust the parameters of the simulator until its outputs look realistic. Our method can work on very detailed simulators which model individual people interacting at specific locations in different locales whereas other methods can only fit very simple simulators.

repository includes network topologies generated based on cellphone geolocation data and county-level census data obtained from SafeGraph; raw data from SafeGraph can be freely obtained for academic use by requesting an account at https://www.safegraph.com/academics. This repository also includes instructions to download county-level COVID-19 infection and death statistics from the COVID-19 Data Repository by the Center for Systems Science and Engineering (CSSE) at Johns Hopkins University (https://github.com/CSSEGISandData/COVID-19).

**Funding:** RW is supported by a Postdoctoral Fellowship from the Roux Institute. JWvdM is also supported by the Intel Corporation, the 3M Corporation, NSF award 1835309, startup funds from Northeastern University, the Air Force Research Laboratory (AFRL), and DARPA. RC is funded by MIT Lincoln Laboratory and the Under Secretary of Defense for Research and Engineering under Air Force Contract No. FA8702-15-D-0001. The funders had no role in study design, data collection and analysis, decision to publish, or preparation of the manuscript.

**Competing interests:** The authors have declared that no competing interests exist.

# 1 Introduction

Planning and control of disease spread often relies on having access to realistic simulation models that can capture fine-grained dynamics and differences between geographic regions. Models of infectious diseases track the number of individuals in different stages of disease progression and with varying granularity. The least granular models track *global* population totals in compartments such as susceptible, infected, and removed [1], and rely on an assumption of uniform mixing such that the simulated population can be fully described by a system of differential equations. This simplifies computation, but also imposes limitations on dynamics, such as the fact, that unlike real-world infection data with multiple waves of infection, this model can only generate a single wave [2]. More sophisticated simulators stratify the population according to age [3] or geography [4] to account for variations in the frequency of interactions. The most fine-grained simulation models are agent-based [5], and can account for the fact that highly-connected individuals are more likely to both contract and spread a disease [6].

Fine-grained models are desirable for epidemiologists who seek to encode important domain knowledge, such as the effects of pre-existing patient states (e.g. age, co-morbidities, or even genetic predispositions) and regional differences in mobility and policy. However, such models will typically have a large number of parameters, and estimating these parameters poses challenges. Some parameters, such as those describing person-to-person interaction frequency, can be estimated prior to simulation using available data on mobility and demographics. Other parameters, such as disease transmission rates, may have substantial uncertainty, particularly during the early stages of an ongoing epidemic, or may be sensitive to implementation details of a simulator so that they cannot be estimated externally. Moreover, parameter estimates based on past data can be invalidated by public health intervention policies affecting mobility and social interaction. In such settings, we would like to deploy models that can incorporate high-resolution data from as many sources as possible. Then, we can apply techniques for maximum likelihood estimation and approximate inference to reason about the most probable parameter values.

In this paper, we present a case study in the use of probabilistic programming methods to infer parameters of agent-based disease simulations. Probabilistic programming research has developed a wide variety of methods for inference in programmatically specified models, including methods based on Markov Chain Monte Carlo (MCMC) [7], importance sampling and variational inference [8]. An example of the use of probabilistic programming in disease modeling is the work by Flaxman et al. [9], which applies methods based on Hamiltonian Monte Carlo (HMC) to comparatively low-granularity compartmental models. HMC is widely used and can be computationally efficient, but requires a differentiable model that defines a density with continuous support. These requirements are not always easy to satisfy. Complex simulations, such as the ones we consider here, may incorporate non-differentiable aspects such as discrete variables or have discontinuities arising from if-then-else statements. Moreover, it is not always convenient to (re-)implement simulations in a special-purpose modeling language that supports automatic differentiation.

Parameter estimation in complex disease simulations typically relies on comparatively simple methods such as importance sampling using likelihood weighting or approximate Bayesian Computation [3, 4]. Approximate Bayesian Computation (ABC) relies on a heuristic likelihood function to compare observed data and predictions from the simulation [10, 11].

However, there are numerous other methods from the probabilistic programming literature that impose few requirements on the underlying model, and can be used to perform inference in simulation-based models in a variety of languages with relatively few changes to the underlying code base. Examples include single-site Metropolis-Hastings [12], importance-sampling

methods based on Sequential Monte Carlo [13], and stochastic variational inference methods [13], which can be combined with deep learning to train models that invert probabilistic programs for fast inference at test time [14, 15].

In order to apply probabilistic programming methods to parameter estimation for fine-grained disease simulations, we implement an agent-based simulation in the Julia language. We model disease transmission on subsampled social proximity networks constructed from cellphone co-location data (Sec. 4.2), in which nodes represent individuals who transition stochastically between disease states (Alg. 4). We infer transmission parameters of this model using Blackbox Variational Inference (BBVI, [16, 17]), a well-established stochastic variational method for probabilistic programs. We use an implementation of BBVI in the Gen probabilistic programming system [18]. We find that this method scales well with the relatively computationally intensive simulations in this model, and outperforms simpler methods based on likelihood-weighted importance sampling and Metropolis-Hastings in terms of sample quality and diversity. To separately capture the effects of changes in mobility from changes in transmission during interaction (e.g. due to non-pharmaceutical interventions like mask-wearing), we include experiments where the contact network varies over time. To explore the sensitivity of our model to the network size, we also include experiments with up- and down-sampled graphs. We find that our model gives high-quality samples in both of these cases. Lastly, we consider an experiment in which we simultaneously fit both infection and death data, and find that our model is able to incorporate both pieces of evidence successfully.

**Limitations**. We make several simplifying assumptions which are common in the literature. These are both in terms of the level of realism of our model design and the quality of our inference approximation.

First, our simulator models the network topology of a region as a subsampled population of nodes. This subsampling necessitates rescaling to compare model outputs to reported case counts, and means that parameter estimates will depend on the degree of subsampling. Our simulator is nonetheless relatively granular compared to other disease models and balances well between speed and resolution.

Second, we perform variational inference for the parameters in our model using a so-called fully-factorized approximation. This approximation makes the optimization problem simpler, since fewer variational parameters have to be learned, but does have well-documented drawbacks. Its main limitation is that a factorized variational distribution cannot capture correlations between parameters in the posterior, which in turn can result in an under-approximation of the posterior variance.

A further limitation is that our gradient estimates have a high variance. The reason for this is that we treat the stochastic simulator as a black box. This means that the distribution over parameters is optimized to maximize agreement with observed data, but simulator trajectories are sampled from a broad prior distribution that is defined in terms of these trajectories (see Fig 1). As a result, the accuracy of the posterior approximation may be limited by the variance of the gradient signal.

Lastly, our method is not designed to model the evolving temporal dynamics of the disease. We are able to fit time-varying infectiousness parameters over a past time range based on data, but our model cannot extrapolate into the future except by assuming these parameters stay constant.

**Contributions**. The present work evaluates the utility of probabilistic programming methods for parameter estimation in simulations that model disease transmission at the level of individuals. For this purpose, we perform a relatively comprehensive set of numerical experiments. Our results demonstrate that even BBVI, a relatively simple simulation-based inference method, can be used for parameter inference in these models. We hope this will inspire further

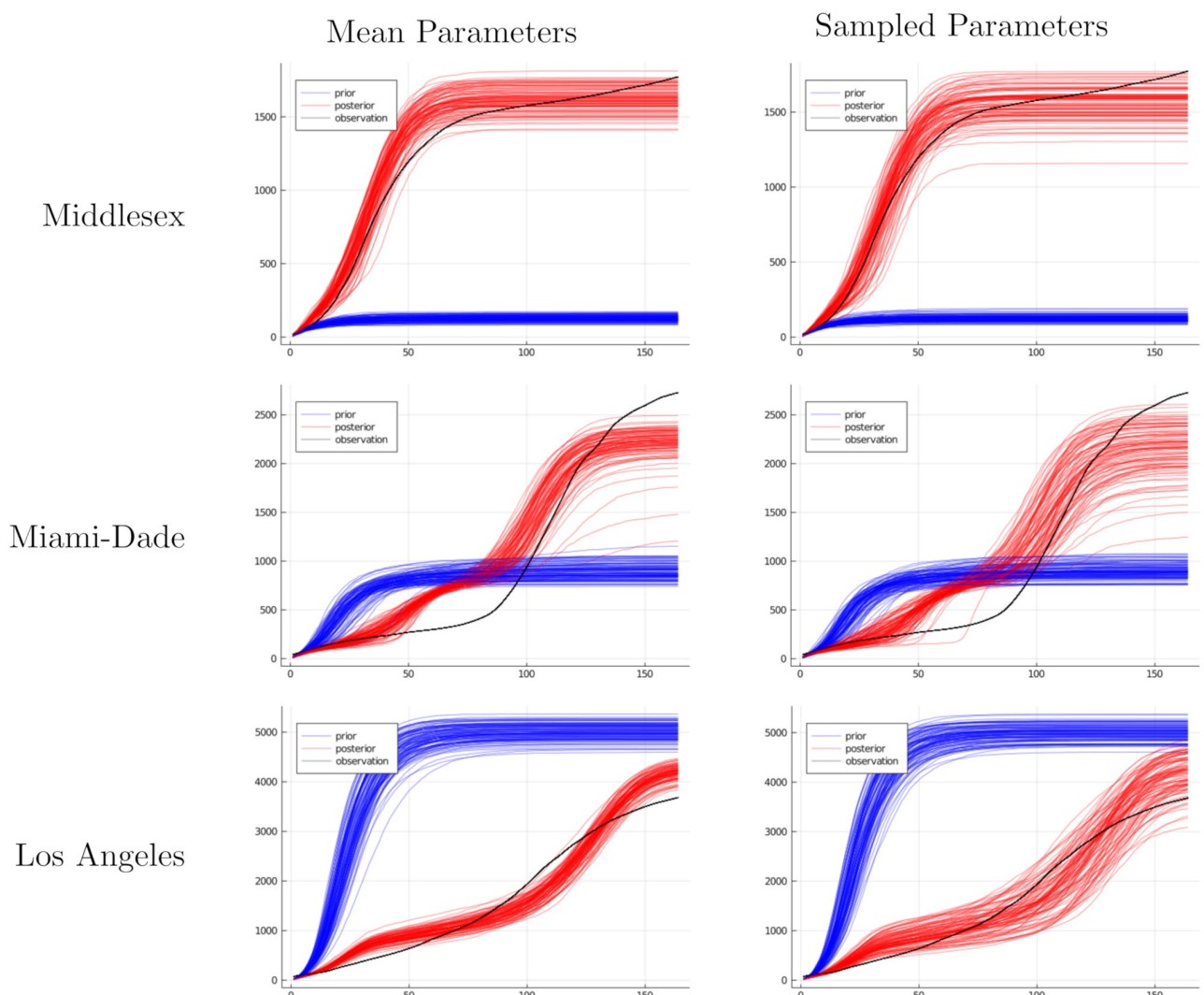

**Fig 1. Cumulative infection trajectories using fixed or sampled disease parameters.** After fitting the parameters of our model to data, we show the cumulative infections produced by 100 runs of the simulator using selected parameters. On the left, we use the posterior mean parameters; the observed variation comes from the untraced randomness of network simulator. On the right, we use samples from the posterior distribution; variation comes from both the untraced randomness and the variance of our posterior distribution.

applications of probabilistic programming to this domain. In particular we observe that variational inference methods make it possible to estimate region-specific parameters in a manner that results in a higher quality of fit when comparing to simpler compartmental models and other inference methods commonly used in related work.

## 2 Background: Inference in simulators

In this paper, we apply methods for Bayesian inference to reason about unknown parameters in disease simulations. Our goal is to approximate a posterior density over latent variables $z$ that is conditioned on observed data $x$,

$$p(z \mid x) \quad = \frac{p(x \mid z)\, p(z)}{p(x)}, \quad p(x) = \int dz \; p(x \mid z)\, p(z). \tag{1}$$

This density is defined in terms of a likelihood $p(x \mid z)$, which reflects the agreement between observed data $x$ and latent variables $z$, and a prior $p(z)$ over latent variables $z$ in the absence of observations. The computational challenge in approximating the posterior is that integrals with respect to $z$ are generally intractable. This means that we cannot compute the marginal likelihood $p(x)$ in closed form and it is generally difficult to approximate the expected value of any quantity $h(z)$ that depends on the latent variables,

$$\mathbb{E}_{z \sim p(z|x)}[h(z)] = \int dz \; p(z \mid x) \, h(z). \tag{2}$$

In this work, observed data takes the form of a time series $x = (x_1, \ldots, x_T)$, where each time point $x_t$ is a count of reported cases. The latent variables $z = \{\theta, s\}$ comprise model parameters $\theta$ and a sequence $s = (s_1, \ldots, s_T)$ in which $s_t$ denotes the state of a disease simulator at time $t$. We express the probability of latent variables and observations as,

$$p(x, z) = p(x, s, \theta) \; = \; p(x \mid s, \theta) \, p(s \mid \theta) \, p(\theta). \tag{3}$$

This model has three components. The first is a likelihood $p(x \mid s, \theta)$, the second is a distribution over disease simulations $p(s \mid \theta)$, and the third is a prior $p(\theta)$.

It is possible to define all model components in a probabilistic programming language to facilitate inference later on. However, this may not be the most convenient choice when applying probabilistic program inference methods to an existing simulation code base, since all simulation code must now be translated or modified to ensure that random variables are generated in a manner that can be tracked by the probabilistic programming framework.

In this paper we will instead focus on inference methods that require no changes to an existing simulation code base. These methods can be applied to a stochastic simulator that is "opaque", which is to say that randomness inside the simulation is not accessible to the inference algorithm. In the Gen probabilistic programming system [18], which we use as the basis for our experiments, these uncontrolled random choices are referred to as "untraced" randomness.

Concretely, we will assume that we have access to a simulator $f$ with inputs $\theta$, and that execution of this simulator generates a random trajectory $s \sim f(\theta)$. This implicitly defines a distribution over trajectories $s \sim p(\cdot \mid \theta)$. However, the density associated with this distribution is intractable, which is to say that we cannot compute $p(s \mid \theta)$. To perform inference in a disease simulation, we will assume that the user implements additional code to define a likelihood $p(x \mid s, \theta)$ and a prior $p(\theta)$,

$$p(x \mid s, \theta) \quad = g(x, s, \theta), \quad p(\theta) = f_0(\theta). \tag{4}$$

We will assume that the prior $f_0$ is a density or probabilistic program that is completely tractable, which is to say that we can sample $\theta \sim f_0$ and evaluate the density $f_0(\theta)$. The likelihood $g$ may take the form of a tractable parametric density, as is the case for the Gaussian likelihood that we define in Section 4.3. Alternatively, we can define a likelihood $g(x, s, \theta) \propto \exp[-\ell(x, s)]$ in terms of a loss function $\ell(x, s)$ that measures the discrepancy between the simulation state $s$ and the observations $x$. Such a loss function should be chosen to ensure that the normalizer $Z = \int dx \exp[-\ell(x, s)]$ is a constant that is independent of $s$. We will discuss an example of such a heuristic likelihood in our discussion of ABC rejection algorithms below.

## 2.1 Likelihood weighting (LW)

Probabilistic programming systems implement general-purpose inference methods that can be applied to simulation-based models. These methods approximate the posterior using samples

that are generated by repeatedly running the simulation. One of the simplest of these methods is importance sampling using likelihood weighting, which generates samples by proposing $\theta$ from the prior, running the simulator to generate a trajectory $s$, and computing an importance weight $w$ according to the likelihood,

$$w^k = g(x, s^k, \theta^k), \quad s^k \sim f(\theta^k), \quad \theta^k \sim f_0, \quad k = 1, \ldots, K. \tag{5}$$

After normalizing these weights by their sum, this method can be used to approximate expectations with respect to the posterior,

$$\mathbb{E}_{s,\theta \sim p(s,\theta|x)}[h(s,\theta)] \simeq \sum_{k=1}^{K} \frac{w^k}{\sum_{k'} w^{k'}} \, h(s^k, \theta^k). \tag{6}$$

A convenient property of this style of inference is that it can be applied to any stochastic simulator. Likelihood weighting does not require evaluation of $p(s \mid \theta)$ at any step in the computation. The only requirements for likelihood weighting is that we should be able sample $\theta$ from the prior, run the simulator to generate a trajectory $s$ and evaluate the user-defined likelihood $g(x, s, \theta)$.

**Algorithm 1**: Likelihood Weighting

```
Function LIKELIHOOD-WEIGHTING(g, f, f₀, K):
    for k ← 1, ..., K do
        θᵏ ~ f₀      // Propose parameters θ
        sᵏ ~ f(θᵏ)      // Run simulator to generate trajectory s
        wᵏ ← g(x, sᵏ, θᵏ)      // Compute unnormalized importance weight w
    return {(wᵏ, sᵏ, θᵏ)}ᵏₖ₌₁      // Return weighted samples
```

## 2.2 Black-box variational inference

One of the limitations of likelihood weighting is that it is not a particularly efficient inference strategy. The prior $f_0$ typically defines a distribution over a broad range of parameters, out of which only a small subset are likely to give rise to simulation trajectories that are in good agreement with the data. That is, likelihood weighting is a form of guess-and-check sampling—it typically requires many proposals $s^k$, $\theta^k$, out of which the overwhelming majority will be in poor agreement with the data. This means that the weighted average in Eq 6 will typically be dominated by a small fraction of high-weight samples.

Variational inference methods directly approximate the posterior by optimizing the parameters of a variational distribution, thereby turning an inference problem into an optimization problem. In this paper, we assume a distribution $q_\phi(\theta)$ with variational parameters $\phi$. If we combine the variational distribution with the simulator $f$, then this defines a variational family $q_\phi(s, \theta)$,

$$q_\phi(s, \theta) = p(s \mid \theta) \, q_\phi(\theta). \tag{7}$$

Our goal is to learn parameters $\phi$ such that $q_\phi(s, \theta)$ is as similar as possible to the posterior $p(s, \theta \mid x)$. Since $q_\phi(s \mid \theta) = p(s \mid \theta)$ is just the distribution over trajectories in the simulator, for which there are no optimizable parameters, making $q_\phi(s, \theta)$ as similar as possible to $p(s, \theta \mid x)$ equates to making $q_\phi(\theta)$ as similar as possible to the posterior marginal over parameters $p(\theta \mid x)$.

To learn the variational parameters $\phi$, we will in this paper use black-box variational inference (BBVI) [16, 17], which is one of the simplest and most widely implemented methods for probabilistic programs. BBVI minimizes the Kullback-Leibler (KL) divergence $\text{KL}(q_\phi(s, \theta) \| p(s,$

$\theta \mid x)$) by maximizing a variational lower bound $\mathcal{L}$ (see S1 Appendix for further discussion),

$$\mathcal{L} \quad = \mathbb{E}_{s,\theta \sim q_\phi}\left[\log \frac{p(x,s,\theta)}{q_\phi(s,\theta)}\right] = \mathbb{E}_{s,\theta \sim q_\phi}\left[\log p(x) + \log \frac{p(s,\theta \mid x)}{q_\phi(s,\theta)}\right] \tag{8}$$

$$= \log p(x) - \mathrm{KL}(q_\phi(s,\theta) \parallel p(s,\theta \mid x)). \tag{9}$$

Since $\log p(x)$ does not depend on the variational parameters $\phi$, maximizing $\mathcal{L}$ with respect to $\phi$ is equivalent to minimizing the KL divergence, which ensures that $q_\phi$ becomes as similar as possible to the posterior.

**Algorithm 2**: Black-box Variational Inference

```
Function BBVI (g, f, f₀, q_φ, K, α_{1:T}):
  for a ← α₁, ..., α_T do
    for k ← 1, ..., K do
      θ^k ~ q_φ       // Propose parameters θ
      s^k ~ f(θ^k)       // Run simulator to generate trajectory s
      w^k ← g(x,s^k,θ^k) f₀(θ^k)/q_φ(θ^k)      // Importance weight w (Eq 12)
    b̂ ← (1/K)∑_{k=1}^{K} log w^k     // Compute control variate
    ĝ ← (1/K)∑_{k=1}^{K}(log w^k − b̂)∇_φ log q_φ(z^k)     // Gradient est. (Eq 12)
    φ ← φ + a ĝ     // Update variational parameters
  return q_φ     // Return variational distribution
```

The variational objective can equivalently be decomposed into an expected log likelihood and a KL divergence between the variational distribution and the prior,

$$\mathcal{L} \quad = \mathbb{E}_{s,\theta \sim q_\phi}\left[\log \frac{p(x,s,\theta)}{q_\phi(s,\theta)}\right] = \mathbb{E}_{s,\theta \sim q_\phi}\left[\log p(x|s,\theta) + \log \frac{p(s,\theta)}{q_\phi(s,\theta)}\right] \tag{10}$$

$$= \mathbb{E}_{s,\theta \sim q_\phi}[\log p(x \mid s,\theta)] - \mathrm{KL}(q_\phi(s,\theta) \parallel p(s,\theta)). \tag{11}$$

The variational objective represents a trade-off between two objectives. The first is to maximize the expected log likelihood, which tends to concentrate the variational distribution $q_\phi(\theta)$ around the parameters $\theta^*$ that yield the highest agreement between simulation and data. The second is to minimize the KL divergence relative to the prior, which tends to make the variational distribution more broadly peaked.

BBVI optimizes the variational objective $\mathcal{L}$ using stochastic gradient ascent, which requires an unbiased estimate of the gradient $\nabla_\phi \mathcal{L}$. To compute this estimate, BBVI uses samples $s^k$, $\theta^k \sim q_\phi$ to compute a likelihood-ratio estimator (see S1 Appendix),

$$\nabla_\phi \mathcal{L} \quad \simeq \frac{1}{K}\sum_{k=1}^{K}\nabla_\phi \log q_\phi(\theta^k)\left(\log w^k - \hat{b}\right), \quad w^k = g(x,s^k,\theta^k)\frac{f_0(\theta^k)}{q_\phi(\theta^k)}. \tag{12}$$

The constant $\hat{b}$, which is known as a baseline, serves to reduce the variance of the estimator. Here, we use a simple baseline in the form of the average log weight.

As with likelihood weighting, BBVI is very broadly applicable to simulation-based models. To compute the weights $w^k$, we need to evaluate the likelihood $g$, the prior $f_0$, and the variational density $q_\phi$. The only additional requirement is that we can compute the log gradient $\nabla_\phi \log q_\phi(\theta)$. As a result, BBVI can be implemented by simply running the simulator $f$ repeatedly to generate samples.

## 2.3 Metropolis-hastings

One of the baseline methods that we use in this paper is single-site Metropolis-Hastings (MH) [12], which is a Markov Chain Monte Carlo (MCMC) method that is implemented in many probabilistic programming systems. In MH, we randomly initialize a sample $\theta \sim f_0$ from the prior. We then apply a series of MCMC updates. For each update, we propose a change $\theta' \sim q(\theta'|\theta)$. The proposed change is then either accepted or rejected according to a Metropolis-Hastings ratio. In the resulting Markov chain, early samples will be representative of the prior distribution, but successive updates define a biased random walk that converges to the posterior.

In the context of simulation-based inference, there is a subtlety to performing MH sampling. The MH acceptance ratio $\alpha$ for a simulation-based model has the form,

$$\alpha = \frac{p(x, \theta')\, q(\theta \mid \theta')}{p(x, \theta)\, q(\theta' \mid \theta)}, \qquad p(x, \theta) = \int ds\, p(x, s, \theta). \tag{13}$$

Unfortunately, we cannot compute this acceptance ratio directly, since doing so requires computing $p(x, \theta)$, which involves an integral with respect to all simulation trajectories $s$. However, we can use likelihood weighting to compute an acceptance ratio by running the simulator to generate trajectories $s' \sim f(\theta')$ and $s \sim f(\theta)$, and computing the acceptance ratio in terms of importance weights,

$$\hat{\alpha} = \frac{w'\, p(\theta')\, q(\theta \mid \theta')}{w\, p(\theta)\, q(\theta' \mid \theta)}, \qquad w = g(x, s, \theta) = p(x|s, \theta) = \frac{p(x, s \mid \theta)}{p(s \mid \theta)}. \tag{14}$$

This amounts to replacing the marginal likelihood $p(x \mid \theta')$ with an unbiased estimate $w'$ and similarly replacing $p(x \mid \theta)$ with an unbiased estimate $w$, since,

$$\mathbb{E}_{s \sim p(s|\theta)}[w] = \int ds\, p(s \mid \theta) \frac{p(x, s \mid \theta)}{p(s \mid \theta)} = p(x \mid \theta). \tag{15}$$

A justification for replacing $\alpha$ with $\hat{\alpha}$ can be derived by noting that the resulting sampler satisfies a technical definition known as proper weighting [19, 20].

Algorithm 3 summarizes the resulting MH sampling procedure when using a proposal that updates a single randomly selected parameter $\theta_i'$, keeping all remaining variables constant $(\theta_{j \neq i}' = \theta_{j \neq i})$ This sampler is once again very general. To generate a proposal and compute the acceptance ratio $\hat{\alpha}$, we only need to be able to sample from the proposal kernel $q(\theta'|\theta)$, run the simulator to generate $s' \sim f(\theta')$, and compute the weight $w' = g(x, s', \theta')$. This construction is once again generally applicable to probabilistic programs that make calls to stochastic functions $f$ whose random choices are opaque to the inference algorithm.

**Algorithm 3**: Single-Site Metropolis Hastings

```
Function TRACE-MH(g, f, f₀, q, K):
  θ ~ f₀    // Initialize from prior
  for k ← 1, ..., K do
    s ~ f(θ); w ← g(x, s, θ)     // Simulate and compute current
weight
    i ~ Uniform({1, ..., |θ|})     // Select site i
    θᵢ' ~ qᵢ(· | θ); θ'ⱼ≠ᵢ ← θⱼ≠ᵢ     // Propose for site i
    s' ~ f(θ'); w' ← g(x, s', θ')     // Simulate, compute proposal
weight
    α̂ ← (w' f₀(θ') qᵢ(θᵢ | θ'))/(w f₀(θ) qᵢ(θᵢ' | θ))     // Compute acceptance ratio (Eq 13)
    u ~ UNIFORM(0, 1)
```

```
        if u < α̂ then  θ, s ← θ′, s′      // Update sample if move is accepted
        θᵏ_out, sᵏ_out ← θ, s       // Store current sample for output
return {θᵏ_out}ᴷ_{k=1}       // Output chain of samples
```

## 2.4 Inference methods for differentiable models

While BBVI does not require the simulator to be differentiable, there are a number of more efficient inference methods that can be used when a model does support differentiation. Examples include variants of Hamiltonian Monte Carlo [21], and reparameterized methods for variational inference [22, 23]. To apply these methods, the model must be implemented in a language that supports automatic differentiation, such as Stan [24], PyMC [25], or Pyro [15]. Moreover, since these methods rely on computation of the gradient of the log density with respect to the latent variables $\nabla_z \log p(x, z)$, all variables in the model need to be continuous. For this reason, support for these methods must be factored into the design of a simulator from the outset, and cannot easily be applied to code bases that do not already support automatic differentiation. Furthermore, these design constraints limit the expressivity of disease simulators, since it is not possible to apply differentiation to models with discrete random variables. By contrast, the BBVI methods that we consider in this paper have fewer implementation requirements, but typically require a much larger amount of computation to approximate the posterior.

## 3 Related work

**Markov chain Monte Carlo methods**. For comparatively low granularity models, such as models with global compartments, it is often possible to apply MCMC methods, which are guaranteed to asymptotically converge to the posterior. Hamiltonian Monte Carlo (HMC, [21, 26, 27]) methods are amongst the most efficient and widely used MCMC methods in this context. An example of the usage of such methods is the work by Flaxman et al. [9] using models implemented in the Stan probabilistic programming language [24]. However, as discussed in Section 2.4, applying HMC requires that the model be differentiable with respect to the latent variables and, practically, the number of needed gradient evaluations be computationally feasible. This makes it difficult to apply HMC to larger-scale disease simulations (on the order of 10000 agents), such as the ones that we consider in this paper. These simulations are not always differentiable, since they may contain discrete random variables, or may simply not be implemented in a language that supports differentiation.

**Likelihood weighting**. Inference in larger-scale simulation-based models has typically relied on much simpler techniques in order to reduce implementation requirements and balance the quality of approximation with computational cost. One of the more commonly used methods in this context is likelihood weighting (see Alg. 1). Work by Wood et al. [28] uses a probabilistic programming implementation of likelihood weighting to perform inference in FRED [5], an agent-based disease simulator. Wilder et al. [29] use likelihood weighting to fit 4 parameters: (1) the probability of infection after contact with an infected individual, (2) the start time of an infection, (3) a base mortality rate multiplier, (4) the reduction factor in expected number of contacts after a lockdown. They select a negative binomial distribution as a likelihood function, where the dispersion parameter is estimated by fitting an autoregressive binomial regression model. Samples are generated approximately from a uniform prior using Latin hypercube sampling.

**ABC rejection algorithms**. Another class of methods related to likelihood weighting are ABC rejection algorithms. These algorithms compare the simulation output $s$ to the data $x$ according to some error function $\epsilon(x, s)$ and reject all samples whose error exceeds a threshold

$\epsilon_0$. This is a special case of likelihood weighting using a heuristic likelihood of the form,

$$g(x, s, \theta) \propto \exp(-\ell(x, s)), \qquad l(x, s) = \begin{cases} 0 & \epsilon(x, s) \leq \epsilon_0, \\ \infty & \epsilon(x, s) > \epsilon_0. \end{cases} \qquad (16)$$

In the limit where $\epsilon_0 \to 0$, this heuristic likelihood conditions the simulation output to exactly match the observations. Increasing the threshold $\epsilon_0$ results in a higher sample efficiency, at the expense of yielding larger approximation errors.

Chinazzi et al. [3] study the effect of travel restrictions on the spread of the coronavirus. They use ABC rejection algorithms to estimate the posterior distribution of the reproductive number $R_0$. Specifically, they choose parameter values from samples which have simulated case counts that match the observed number of cumulative imported cases before January 23, 2020 to within a margin of error of +40%.

Similarly, Chang et al. [4] use ABC to fit model parameters to the number of confirmed cases provided by the New York Times. They estimate 3 parameters: (1) base transmission rate, (2) point-of-interest transmission rate, (3) initial proportion of exposed individuals. These parameters are fit by performing a grid search, where the utility of parameters is computed based on the average root mean squared error (RMSE) over 30 random simulations. To quantify uncertainty, the authors select parameters whose average RMSE is within 20% of the best-fitting parameter set, and report the mean and 2.5–97.5th percentile range of parameter values.

Unlike the ABC method used by Chinaazi et al. and Chang et al., our method does not use a hard threshold for rejecting samples, which increases sample efficiency.

## 4 Methods

In this work, we consider a disease simulation model that we call Network-SEIR (NSEIR), which comprises two components. The first is a network topology model, in the form of a degree-corrected stochastic block model (DCSBM [30]). This model describes contact patterns in the population. We obtain point estimates of network topology parameters using cell-phone co-location data [31] in order to produce a representative graph for simulations. The second component is an agent-based compartmental model that describes how disease spreads across this network topology to produce simulated infection statistics.

To calibrate our Network-SEIR model to a particular region, we seek to learn input parameters to this simulator that produce infection statistics matching the region's true outcomes. We incorporate this agent-based model into a probabilistic program by defining a prior over these input parameters and a likelihood for reported case counts. The resulting probabilistic program defines a Bayesian posterior over parameters that we approximate using variational inference.

### 4.1 Compartmental SEIR models

We begin by briefly reviewing traditional global compartmental SEIR models in order to motivate the agent dynamics in our Network-SEIR model. In a SEIR model, the population is separated into four compartments representing Susceptible ($S$), Exposed ($E$), Infected ($I$), and Removed ($R$) individuals. These compartments are approximated by continuous values $S(t)$, $E(t)$, $I(t)$, and $R(t)$, where the total population size is fixed at $N = S(t) + E(t) + I(t) + R(t)$. The

population dynamics are modeled by the differential equations,

$$\frac{dS}{dt} = -\frac{\beta IS}{N}, \qquad \frac{dE}{dt} = \frac{\beta IS}{N} - \gamma E, \qquad \frac{dI}{dt} = \gamma E - \lambda I, \qquad \frac{dR}{dt} = \lambda I. \tag{17}$$

In some settings, this model includes natural rates of birth and death for the population; we omit these factors from our network model for two reasons. First, newborn individuals will not contribute meaningfully to the spread of infection on their own; rather, they can be treated as a sub-compartment of their caregivers. Second, individuals who die of natural causes make the overall mobility network slightly more sparse, and omitting this effect causes only a small error in our predictions.

## 4.2 Network-SEIR model

We now describe the stages of constructing our stochastic disease simulator.

**4.2.1 Network modeling.** The first stage of constructing our disease simulator involves fitting a network model to regional mobility data. Our network model belongs to the popular category of mobility networks called spatial meta-population models [3, 32, 33]. We use mobility data from SafeGraph [31], consisting of opt-in, anonymized foot-traffic to points-of-interest (POIs) such as stores and schools across the United States. Devices in this data are assigned to a home Census Block Group (CBG), the smallest geographical unit for which population data is reported in the U.S. Census. For each region of interest, we collect one week of data beginning on February 17, 2020, shortly before widespread quarantines were instituted in the United States and elsewhere. We then select CBGs that contribute to the top 10% of mobility data available for each county during this time period.

We use an extension of a Degree-Corrected Stochastic Block Model (DCSBM) to generate a synthetic contact network that captures the community structure and heterogeneous degree distribution of real networks. The DCSBM has two parameters: 1) a partition of vertices $\{\mathcal{V}_1, \ldots, \mathcal{V}_c\}$ into $C$ communities, and 2) a symmetric matrix $P \in \mathbb{R}^{C \times C}$ of edge probabilities, where element $P_{rs}$ gives the probability of an edge existing between any two vertices $u \in \mathcal{V}_r$ and $v \in \mathcal{V}_s$. For our model, we apply a degree correction procedure within each community and overlay edges that represent household interactions. We leverage census data to extract the household size distribution for each community.

We choose the number of communities in the modeled network by analyzing convergence properties of core topological properties like network density and number of triangles. The sizes of each community and the density of contact patterns within and across communities are selected to match census and Safe graph data from the corresponding Census Block Groups.

**Edge probabilities**. To construct the block matrix $P$ describing community structure, we compute each entry $P_{rs}$ from the cross-correlation score of the POI visit vectors of $CBG_r$ and $CBG_s$. Let $L \in \mathbb{R}^{C \times N}$ be the visit count matrix for a specific week of data, where $N$ is the number of POIs. The cross-correlation score is given by,

$$P_{rs} = \frac{\sum_{i=1}^{N}(L_{ri} - \bar{L}_r)(L_{si} - \bar{L}_s)}{\sqrt{\sum_{i=1}^{N}\left(L_{ri} - \bar{L}_r\right)^2 \sum_{i=1}^{N}\left(L_{si} - \bar{L}_s\right)^2}}, \quad \bar{L}_r = \frac{1}{N}\sum_{i=1}^{N} L_{ri}. \tag{18}$$

We perform a degree-correction procedure on this network to yield a heterogeneous degree distribution between nodes in different communities. The parameter for degree-correction is sampled from a power-law function with exponent $\gamma = 3$ selected so the node with the largest degree in each community has a degree in the range of 50–100. This is done to be consistent

with other realistic social contact networks of similar sizes [34, 35, 36]. The correction procedure is done on a per-community basis, such that the degree sequence sampling takes into account the size and average degree of each CBG.

This gives us a DCSBM where the edge probability between two CBGs reflects how often individuals from those communities tend to visit the same POIs. From this DCSBM, we sample a network instance $G$ with vertices $\mathcal{V}$ and edges $\mathcal{E}$. Each vertex $v \in \mathcal{V}$ represents a simulated person, and belongs to some $\text{CBG}_i$ (where $i \in \{1, \ldots, C\}$). Vertices are also randomly assigned to households (cliques) within each CBG based on census survey data describing the size distribution of households with 1 to 7 people, resulting in additional "household" edges [37].

**Edge weights**. Given the edge structure described above, we assign edge weights by considering visit overlap duration within a chosen 1-week time window. The weight of edge $(u, v)$ represents the total expected minutes for which individuals $u$ and $v$ might overlap at various POIs. For each POI $p$, SafeGraph provides the median visit duration $d_p$. We build a visit duration tensor $D \in \mathbb{R}^{C \times N \times M}$, where $C$ is the number of CBGs, $N$ is the number of POIs, and $M$ is the total number of minutes POIs are open (we assume $M = 10$ hours per day, 7 days per week = 4, 200 minutes). Specifically, for each POI $p$, we examine our visit count matrix $L$, and include $L_{rp}$ total visits from $\text{CBG}_r$, letting each visit start at a uniform random minute in the interval $(0, M - d_p)$ and last for $d_p$ minutes. An entry $D_{rpt}$ indicates the number of visitors from $\text{CBG}_r$ present at POI $p$ in minute $t$. The weight assigned to an edge between nodes $u \in \mathcal{V}_r$ and $v \in \mathcal{V}_s$ is then given by the total minutes of overlap,

$$W_{uv} = \sum_{t \in (0, M-d_p)\ p \in [1,N]} D_{rpt} D_{spt}. \tag{19}$$

The weight on the additional "household" edges is set to correspond to 8 hours per day (3, 360 weekly minutes).

**4.2.2 Disease transmission model.** Given a representative network instance as described above, we construct a stochastic disease state model as follows. Nodes transition between four states as in the traditional SEIR model: susceptible, exposed, infected, and removed. Let $S_t$, $E_t$, $I_t$, and $R_t$ refer to the subset of nodes in each state at time $t$. We model exposure probability using an exponential distribution,

$$p(v \in E_{t+1} \mid v \in S_t) \quad = 1 - \exp(-E_t^{\text{pressure}} - I_t^{\text{pressure}}). \tag{20}$$

Here the terms $E_t^{\text{pressure}}$ and $I_t^{\text{pressure}}$ affecting node $v$'s transition probability are defined based on its network weights to exposed neighbors $N_t^E(v)$ and infected neighbors $N_t^I(v)$, and scaled by time-dependent transmission parameters $\beta_t^E$ and $\beta_t^I$,

$$E_t^{\text{pressure}} = \sum_{u \in N_t^E(v)} W_{uv}\, \beta_t^E, \qquad I_t^{\text{pressure}} = \sum_{u \in N_t^I(v)} W_{uv}\, \beta_t^I. \tag{21}$$

Once individuals are exposed, they transition to infected and removed states with constant daily probabilities $\gamma$ and $\lambda$,

$$p(v \in I_{t+1} \mid v \in E_t) = \gamma, \qquad p(v \in R_{t+1} \mid v \in I_t) = \lambda. \tag{22}$$

The full disease simulation procedure, which we refer to as $f_{\text{NSEIR}}$, is described in Algorithm 4. The inputs to this model are the simulated regional network $G$ (described by its vertices $\mathcal{V}$ and weighted edges $W$), initial rates of exposure $\rho_c$ in each community, state transition parameters $\gamma$ and $\lambda$, and values for $\beta_n^E$ and $\beta_n^I$ at $N$ time points $\tau_n$, from which we define parameters at all other times $t$ using linear interpolation.

**Algorithm 4**: Stochastic Disease Simulator $f_{\text{NSEIR}}$

```
Function f_NSEIR (G = (V, W), ρ, β^E, β^I, γ, λ, τ, T):
  for c ← 1 to C do     // Initial exposure
    for v ∈ V_c do if UNIFORM(0, 1) < ρ_c then v → E_1 else v → S_1
  for t ← 1, ..., T - 1 do     // Simulate T days
    β_t^E ← INTERPOLATE((τ_1, β_1^E), ..., (τ_N, β_N^E))
    β_t^I ← INTERPOLATE((τ_1, β_1^I), ..., (τ_N, β_N^I)))
    for v ∈ S_t do     // New exposures
      E^pressure ← ∑_{u∈N_t^E(v)} W_uv β_t^E
      I^pressure ← ∑_{u∈N_t^I(v)} W_uv β_t^I
      if UNIF(0, 1) < 1 - exp(-E^pressure - I^pressure) then v → E_{t+1}
    for v ∈ E_t do if UNIF(0,1) < γ then v → I_{t+1}     // Symptoms begin
    for v ∈ I_t do if UNIF(0,1) < λ then v → R_{t+1}     // Infection ends
  return {∑_{t=1}^j |I_t ∩ E_{t-1}|}_{j=1}^T     // List of Cumulative Infections
```

**Linearly approximated transitions**. The exponential transition formula (Eq 20) can be linearly approximated $p(v \in E_{t+1} \mid v \in S_t) \approx E_t^{\text{pressure}} + I_t^{\text{pressure}}$. When using this approximation, we call our disease model Network-SEIR-Linear (NSEIR-Linear). In this approximation, the values and meanings of the edge weights $W_{uv}$ and disease parameters $\beta_t^E, \beta_t^I$ differ relative to the regular NSEIR model. We use the NSEIR-Linear model in our experiments with synthetic data since the $\beta_t^E, \beta_t^I$ values are more easily interpretable; in Eq 20, increasing $\beta_t^E, \beta_t^I$ beyond a certain magnitude has a very small impact on the probabilities due to exponential decay.

## 4.3 Parameter inference

Given a regional network $\mathcal{G}$ and the stochastic disease simulator $f_{\text{NSEIR}}$ from Algorithm 4, we seek to fit the remaining free parameters of our simulator to regional data. To do so, we use BBVI (Section 2.2), as implemented in the Gen probabilistic programming system, to approximate the posterior over model parameters and initial conditions using a variational distribution.

In the $f_{\text{NSEIR}}$ disease simulator, the set of parameters that we will infer is $\theta = \{\rho, \beta^E, \beta^I\}$. We treat the other inputs to the model as fixed hyperparameters, including the latency parameter $\gamma$ = 0.143, and the recovery parameter $\lambda$ = 0.072, (corresponding to a mean latency of 7 days and mean recovery time of 14 days) which we base on clinical case studies [38]. Given these inputs, the simulator returns a sampled trajectory $s$ that contains the number of infected nodes at each time,

$$s_{1:T} \sim f_{\text{NSEIR}}(G, \theta, \gamma, \lambda, \tau, T). \tag{23}$$

To define an inference problem for $f_{\text{NSEIR}}$ simulator, we need to define a prior $f_0(\theta)$, a likelihood $g(x, s, \theta)$ of reported case counts $x$, and a variational distribution $q_\phi(\theta)$.

**Prior**. We define a prior distribution over input parameters to our simulator. We choose to factor this distribution into the product of independent logistic normal distributions over each disease parameter,

$$\tilde{\rho}_c \sim \text{LN}(\hat{\mu}_c^\rho, \hat{\sigma}^\rho), \qquad \rho_c = \frac{\tilde{\rho}_c}{\sum_c \tilde{\rho}_c} E_0 \qquad \text{for} \quad c \in \{1, \dots C\},$$

$$\beta_{t_n}^E \sim \text{LN}(\hat{\mu}_n^E, \hat{\sigma}^E), \qquad \beta_{t_n}^I \sim \text{LN}(\hat{\mu}_n^I, \hat{\sigma}^I) \qquad \text{for} \quad n \in \{1, \dots N\}. \tag{24}$$

Note that there is some flexibility in deciding how to parametrize each component of the variational model. Our first constraint is that the sampled values of each of these variables ($\rho_c$, $\beta_{t_n}^E$, and $\beta_{t_n}^I$) should be non-negative; this suggests the use of log-normal or logistic normal

distributions as relatively standard choices. Since $\rho_c$ represents a percentage of exposed individuals in a community, this parameter should also be restricted in [0, 1]. We similarly chose to restrict the values of $\beta_{t_n}^E$ and $\beta_{t_n}^I$ to the range of [0, 1], such that a node's maximal contribution to the infection of a neighbor is limited by the edge weight between them. The logistic normal distribution becomes a natural choice, as it enforces the desired constraints on these variables. The logistic normal distribution is defined so that values $z \sim LN(\cdot)$ sampled from the distribution can be transformed by the sigmoid function $y = \text{SIGMOID}(z)$ to produce Gaussian-distributed outputs. The parameters defining the prior $\hat{\mu}^\rho, \hat{\sigma}^\rho, E_0, \hat{\mu}^E, \hat{\sigma}^E, \hat{\mu}^I, \hat{\sigma}^I$ are hyperparameters of model tuned with grid search (See Sec. 5.1 and S3 Appendix). We fix the percent exposed on the first day $E_0$ to remove ambiguity between having high $\rho_c$ and low $\beta_{t_0}^E$ versus low $\rho_c$ and high $\beta_{t_0}^E$, which has only a small impact on the likelihood.

**Likelihood**. We define a likelihood that compares the model output $s_{1:T}$ to the reported case counts $x_{1:T}$ for a particular region. Here, we must account for the fact that our agent-based simulator uses a subsampled population that is orders of magnitude smaller than the actual regional population. For this purpose, we define a likelihood model with time-dependent Gaussian noise, which incorporates a scaling factor $r$ to account for the ratio between individuals in the population and nodes in the network topology and a hyperparameter $v$ describing the noise in our observations (see Sec. 5.1 and S3 Appendix),

$$x_t \sim \mathcal{N}(r\,s_t, \; r\,\sigma^x(G, v, t)), \qquad \sigma^x(G, v, t) = v\sqrt{t}|\mathcal{V}|. \tag{25}$$

The above time dependent function $\sigma^x$ was arrived at through experimentation (see Sec. 5.1).

**Variational distribution**. To approximate the model posterior over latent variables, we define a variational distribution $q_\phi(\theta)$ which mirrors the prior of the generative model, with parameters $\phi = \{\mu^\rho, \sigma^\rho, \mu^E, \sigma^E, \mu^I, \sigma^I\}$ for the individual logistic-normal distributions. This results in a fully-factorized variational approximation of the form,

$$q_\phi(\theta) = \prod_{c=1}^{C} q(\rho_c; \mu_c^\rho, \sigma^\rho) \prod_{n=1}^{N} q(\beta_n^E; \mu_n^E, \sigma^E) \, q(\beta_n^I; \mu_n^I, \sigma^I). \tag{26}$$

Note that we share the variance parameters $\sigma_\rho$ across communities, and share variance parameters $\sigma^E$ and $\sigma^I$ across time points. This modeling choice reduces the number of parameters in our model at a small cost to expressivity.

# 5 Results

Our experiments evaluate the extent to which standard probabilistic programming methods, which have been implemented in a wide range of probabilistic programming systems, can be used to estimate parameters in the Network-SEIR model, a representative of agent-based models that are on the computationally intensive end of the spectrum. In particular, we examine whether our approach results in an approximation to the posterior that more accurately mirrors real-world disease spread, especially when compared to other commonly-used methods that assume simplified disease models or fitting techniques.

We first validate the self-consistency and accuracy of our inference model. In Section 5.2, we show that our fitting procedure is well-calibrated by checking if our method is able to infer disease parameters when applied to synthetic data generated by our own simulator using known ground truth values. In Sections 5.3 and 5.4, we compare our method to other methods which use either simplified disease models or simplified fitting procedures common in the

literature. We show our method is able to more accurately fit the complex dynamics of real-world data.

Next, we perform a series of experiments to understand how the output of our inference procedure varies as we vary the regional network topology and regional disease statistics. In Section 5.5, we compare the inferred disease parameters found by applying our method to different regional networks and infection data. In Section 5.6, we explore the pattern of initial disease spread inferred by our model.

Lastly, in Section 5.7, we perform sensitivity analysis for our inference procedure to choices made when modeling the network topologies.

## 5.1 Experiment setup

Before describing the results of our experiments, we first describe data preprocessing, evaluation metrics, and the hyper-parameter tuning procedure.

**Regional infection data processing**. We fit the parameters of our disease model to a particular geographic region by conditioning the model on corresponding county's cumulative infection counts from the Johns Hopkins University Center for Systems Science and Engineering (JHU CSSE) dashboard [39]. We use data from February 29, 2020 until August 9, 2020. We preprocess this data by applying a 7-day rolling average to mitigate the effects of delayed reporting and weekly variation. We then select a 163-day window of investigation, beginning one average latency period ($1/\gamma$ days) before the community infection count reached the chosen initial percentage of exposed individuals $E_0$.

We then rescale the infection counts to the number of nodes in our network topology. Instead of rescaling proportionately by $\frac{\text{nodes}}{\text{county pop.}}$, we scale by a constant $r$ such that the total infection count reaches approximately 50% of the nodes by the end of our simulation time-window (see Eq 25). We do this for two reasons. First, when working with heavily down-sampled graphs, and especially in areas or time periods with limited test information, the number of infections may be so low that the signal is difficult to resolve in simulation. Second, studies have shown a systematic under-counting of cases, for example by comparing the rates of positive PCR tests (which can detect active or recently cleared infection) to the rates of positive serum antibody tests (which can detect historical infection in individuals who were asymptomatic and may not have received a PCR test) [40].

More sophisticated approaches are certainly possible, such as using counts of hospitalizations and deaths (since severe and fatal cases may be less prone to under-counting problems), and then estimating a static or a time-varying fraction of severe and fatal cases. Likewise, it would be possible to model a daily latent variable representing testing rates, which would require conditioning the model on data describing the number of tests administered in a certain region. In this case study, we are primarily interested in evaluating the feasibility of applying probabilistic program inference methods, and we will therefore leave further refinement of the regional data model to future work.

**Evaluation metric**. To evaluate quality of fit for a given disease model, either with fixed disease parameters or a distribution over disease parameters, we generate multiple trajectories from our stochastic disease model, sampling from the disease parameter distribution if necessary. We then compute the mean daily absolute error (MDAE),

$$\text{MDAE}(x, \hat{x}) \quad \equiv \underset{q_\phi(z)}{\mathbb{E}} \left[ \frac{\| f_{\text{NSEIR}}(z) - x \|_1}{TN} \right] \approx \frac{1}{NT} \sum_n \sum_t \frac{1}{S} |f_{\text{NSEIR}}(z^n)_t - x_t|, \qquad (27)$$

where $x$ represents the true cumulative case counts, $\hat{x}$ represents our inferred cumulative case counts using sampled parameters $z$, $N$ represents the number of trajectories computed, $T$

represents the time duration of each trajectory, and $S$ represents the size of the regional population being simulated. MDAE measures the time- and population-normalized distance between the generated cumulative infection counts and true data to which the model was fit.

We evaluate model quality in terms of infection counts rather than inferred parameters because our inference model is conditioned only on infection counts, and there exist multiple solutions for our overparametrized model whose output infection counts would be of similar quality. In the case of experiments on data generated from our own disease model, it is meaningful to compare the disease parameters used to generate the data with those we infer. However, for experiments using real-world data, there are no a priori ground truth values for these disease parameters to compare our learned values to, since these disease parameters are intrinsic to our specific disease model.

Though values for $\beta^E$ and $\beta^I$ are reported elsewhere in the literature, these are, in fact, average transmission rates under the homogenous network assumption valid only for a global compartmental model. In our model, we relax this assumption to take into account the heterogeneity of the network, and thus we cannot directly compare.

**Hyperparameter optimization**. We perform a grid search over hyperparameters for our probabilistic model. This includes hyperparameters such as learning rate and samples per gradient step for the BBVI algorithm, assumed observational noise, and the values for the prior over disease parameters. See Table A in S3 Appendix for the explored parameter ranges.

**Likelihood noise model**. All experiments presented in this paper use the time-dependent noise model $\sigma^x$ described in 4.3. We also considered likelihood models with noise scaling based on the value,

$$\sigma^x_{\text{value-dependent}}(G, x, t) = vx_t|\mathcal{V}|. \tag{28}$$

as well as constant or piecewise constant noise functions,

$$\sigma^x_{\text{const.}}(t) \quad = v, \tag{29}$$

$$\sigma^x_{\text{piecewise}}(t) \quad = \begin{cases} v_0, & \text{if } 0 \leq t \leq \tau_1 \\ \dots \\ v_N, & \text{if } \tau_{N-1} \leq t \leq \tau_N. \end{cases} \tag{30}$$

These alternative approaches did not work as well in early experiments and we did not pursue them further.

## 5.2 Validation on simulated data

To confirm our inference procedure is well-calibrated, we perform inference conditioned on synthetic data. We generate simulated infection counts from our own disease model using known, fixed disease parameters. We then check that BBVI can recover the known disease parameters from the cumulative infection counts alone. For this experiment, we use Network-SEIR-Linear (see Section 4.2.2), a disease model that replaces the exponential transition formula (Eq 20) of Network-SEIR with its linear approximation since Eq 20 is not highly sensitive to changes in disease parameters $\beta^E_t$ and $\beta^I_t$ for higher values of $\beta^E_t$ and $\beta^I_t$.

We perform this experiment by generating data using 6 different time-varying patterns for $\beta^E_t$. For example, the pattern "low-high-low", corresponds to a time interval with low $\beta^E_t$ values, followed by a time interval with high $\beta^E_t$ values and ending with a time interval with low $\beta^E_t$ values. Then, we run our inference procedure and compare trajectories from our model after

**Table 1. MDAE using synthetic data from different counties and disease dynamics.**

| County | low | high | low-high | high-low | low-high-low | high-low-high |
|---|---|---|---|---|---|---|
| Miami-Dade | 0.0052 | 0.0046 | 0.0042 | 0.0051 | 0.0043 | 0.0050 |
| Los Angeles | 0.0037 | 0.0046 | 0.0050 | 0.0044 | 0.0048 | 0.0047 |

fitting to trajectories from the ground truth disease hyperparameters. We find that our method is able to obtain good fits in all cases as shown in Table 1 and Fig 2.

We also compare the inferred $\beta_t^E$ parameter values to those used to generate the plots. In Fig 3, we see that when the high value for $\beta$ was used to generate data, the inferred value was higher and similarly the inferred value was lower when the generating value was lower. This effect held across different scenarios and across time within each scenario. We observe the inferred values for $\beta^E$ do not match the generating values at the end of the simulation. Since the training signal for $\beta^E$ comes from a delayed statistic (cumulative infection counts), the model has little information to constrain the parameter at the end of simulation, and it reverts towards the prior mean of $\beta^E = 0.2$. Furthermore, the model shows some under-fitting due to our choice of large observational noise.

Note that this setting also allows us to evaluate our model in the absence of model misspecification error; whereas real-world infection counts arise from a different system and may or may not be in the range of a particular model, the synthetic data we generate here is known to be in the range of our generative model.

## 5.3 Network versus compartmental disease model

Compartmental disease models are popular in the literature as they are lightweight, deterministic, and differentiable. There are thus many effective strategies for fitting compartmental models to data. Here we use a five-compartment SEIRD model, where $R$ represents recovered and $D$ represents deceased individuals, and fit the model to data using Certainty-Equivalent Expectation Maximization (CE-EM) [41]. To compare with our method, we combine the contents of the Recovered and Deceased compartments to approximate our Removed compartment.

In Table 2, we see that our method achieves better fit across multiple regions. Compartmental models lack expressivity and flexibility and are not able to capture the wide range of disease dynamics we see in different regions. In particular, for fitting data containing phenomena such as multi-wave infections and variable spread-rate resulting from network structure, our agent-based or network-based model achieves much lower MDAE than the compartmental baseline.

## 5.4 Comparing inference methods for network model

We compare BBVI against several alternative procedures for fitting our NSEIR model to data. These alternate methods are used elsewhere in the literature for fitting complex disease models to data. However, due to the large number of parameters in our model and the high degree of untraced randomness, we find these simpler alternative procedures are not able to obtain good fits for Network SEIR.

The alternate fitting procedures we use are (1) $R_t$-analytic (see S2 Appendix), (2) likelihood-weighted importance sampling (IS, Algorithm 1), and (3) Metropolis-Hastings (MH, Algorithm 3). The MDAEs for these methods are listed in Table 2. Samples from NSEIR using parameters fit with these methods are shown in Figs 4 and 5. In order to provide a fair comparison between different inference strategies for fitting the NSEIR model to regional infection

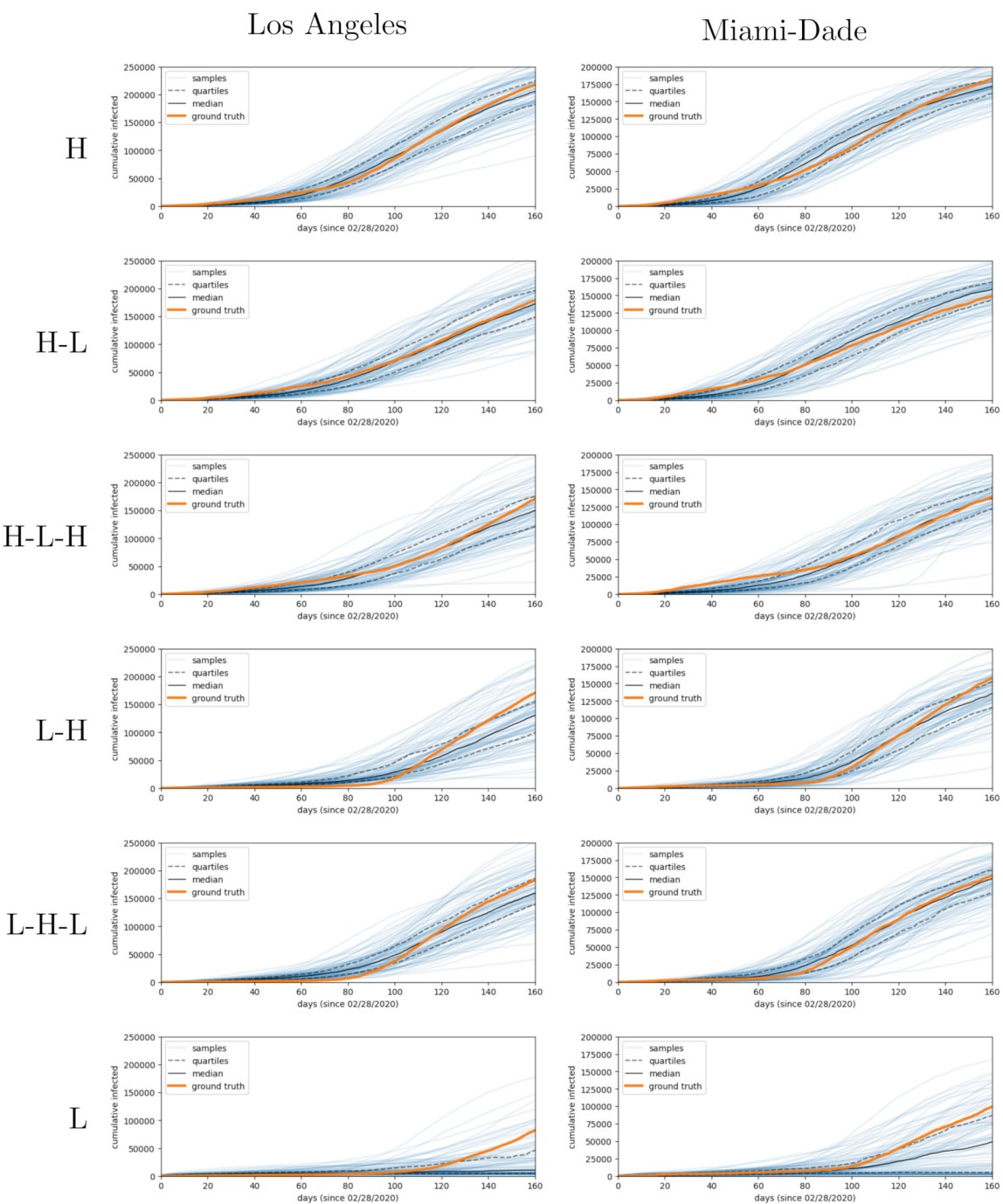

**Fig 2. Sampled infection trajectories after fitting parameters on synthetic data.** We generate simulated data on Los Angeles and Miami-Dade topologies using known disease parameters, and use this data for parameter inference. Generated disease trajectories use "high", "high-low", "high-low-high", "low-high", "low-high-low", "low" patterns, where data is simulated with $\beta^E$ that varies temporally between "high" ($\beta^E = 0.45$) and "low" ($\beta^E = 0.1$) states.

statistics, we use the same prior distribution and run each method with the same total sampling budget of 4800 forward simulations. For BBVI, this sampling is allocated into 120 samples per gradient step, for 40 gradient steps. For MH, we increase sample diversity by running 100 sampling chains independently, each for 480 steps, and retaining only the final sample

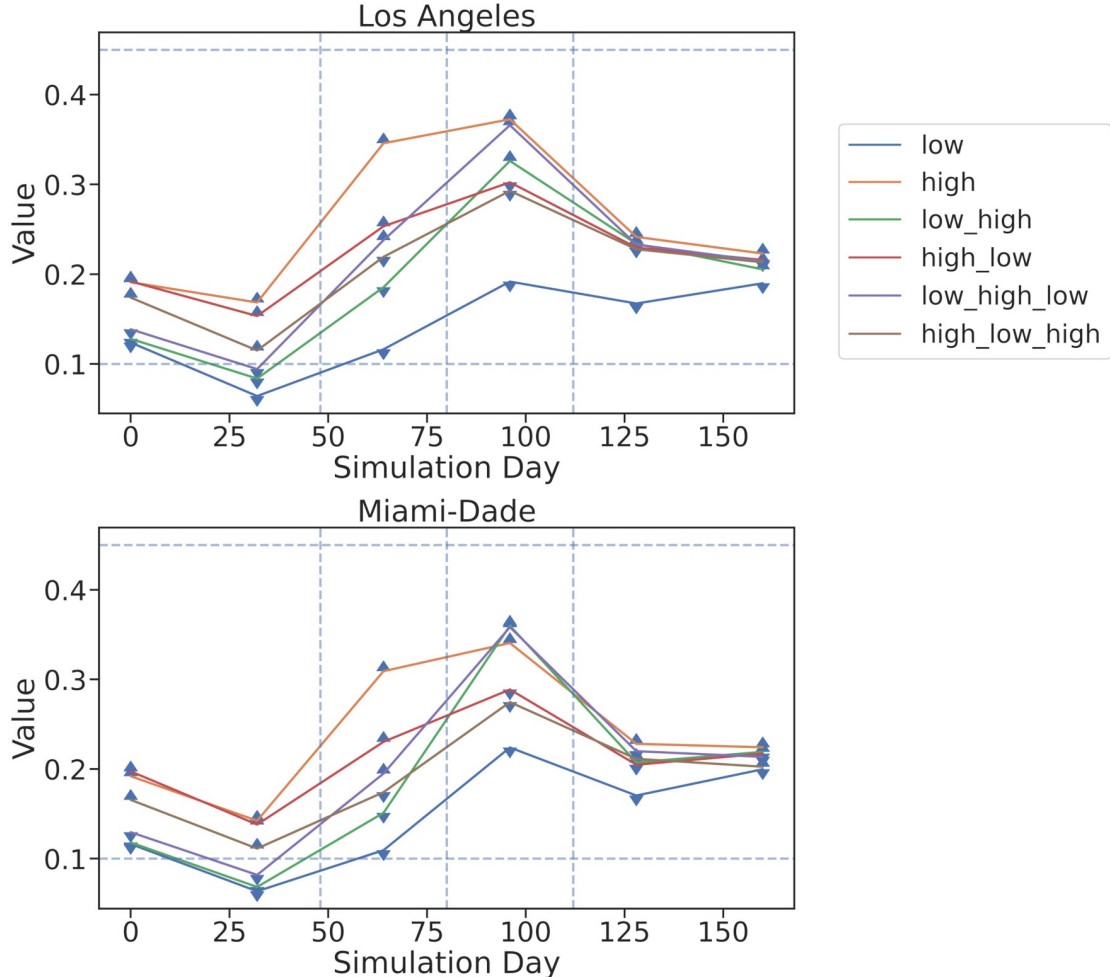

**Fig 3. Inferred parameter values from synthetic data.** We plot the inferred values of $\beta_t^E$ across 6 different generated scenarios using 6 lines. The scenarios are "high", "high-low", "high-low-high", "low-high", "low-high -low", and "low" where $\beta^E$ varies temporally between "high" $\beta^E = 0.45$ and "low" $\beta^E = 0.1$ states, represented by horizontal dotted lines. The vertical dotted lines represents the times when the true parameters were changed while generating data. The value of $\beta$ used when generating the data is indicated by marker with up arrows indicating high and down arrows indicating low. We see that when the high value for $\beta$ was used to generate data, the inferred value was higher and similarly the inferred value was low when the generating value was low. The inferred value for $\beta^E$ is closer to the prior value of 0.2 in all scenarios at the end of the simulation when the signal from the cumulative infection counts is weaker.

**Table 2. Mean daily absolute error (MDAE, Eq 27) for various disease models and fitting procedures across several counties.** BBVI fits the NSEIR model to real infection statistics better than Metropolis-Hastings or Likelihood-weighted importance sampling. NSEIR also outperforms Compartmental SEIR model with CE-EM.

| Disease Model | Fitting Method | Los Angeles | Miami-Dade | Middlesex |
|---|---|---|---|---|
| Compartmental SEIR | CE-EM | 0.0127 | 0.0217 | 0.0080 |
| Network SEIR | $R_t$-analytic | 0.0103 | 0.0367 | 0.0021 |
| Network SEIR | Metropolis Hastings | 0.0124 | 0.0134 | 0.0076 |
| Network SEIR | Likelihood Weighting | 0.0066 | 0.0090 | 0.0056 |
| Network SEIR | BBVI | **0.0011** | **0.0036** | **0.0012** |

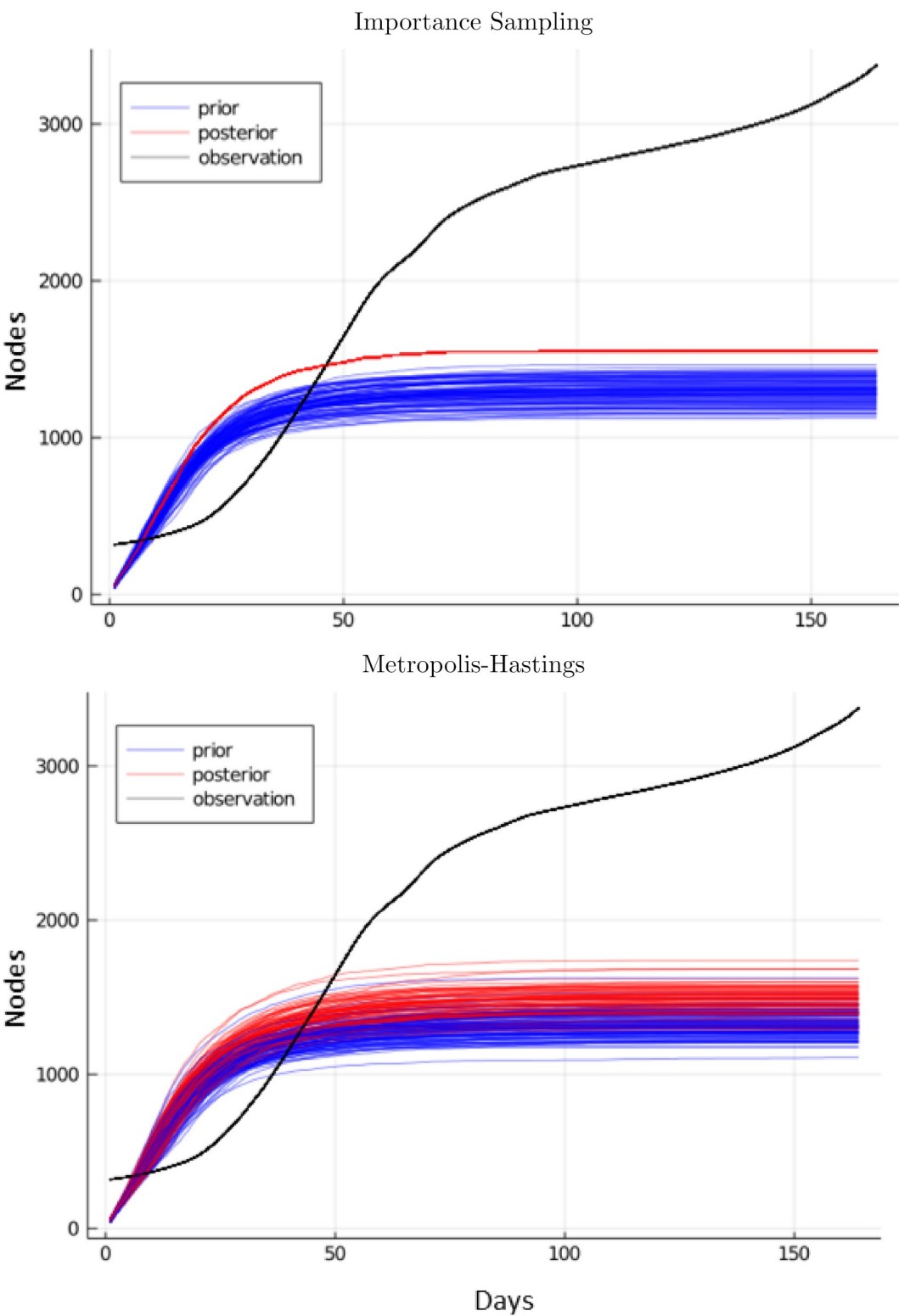

**Fig 4. Samples for the NSEIR model for Miami-Dade using parameters learned from likelihood weighting and Metropolis-Hastings.** Neither alternate method is able to produce a good fit.

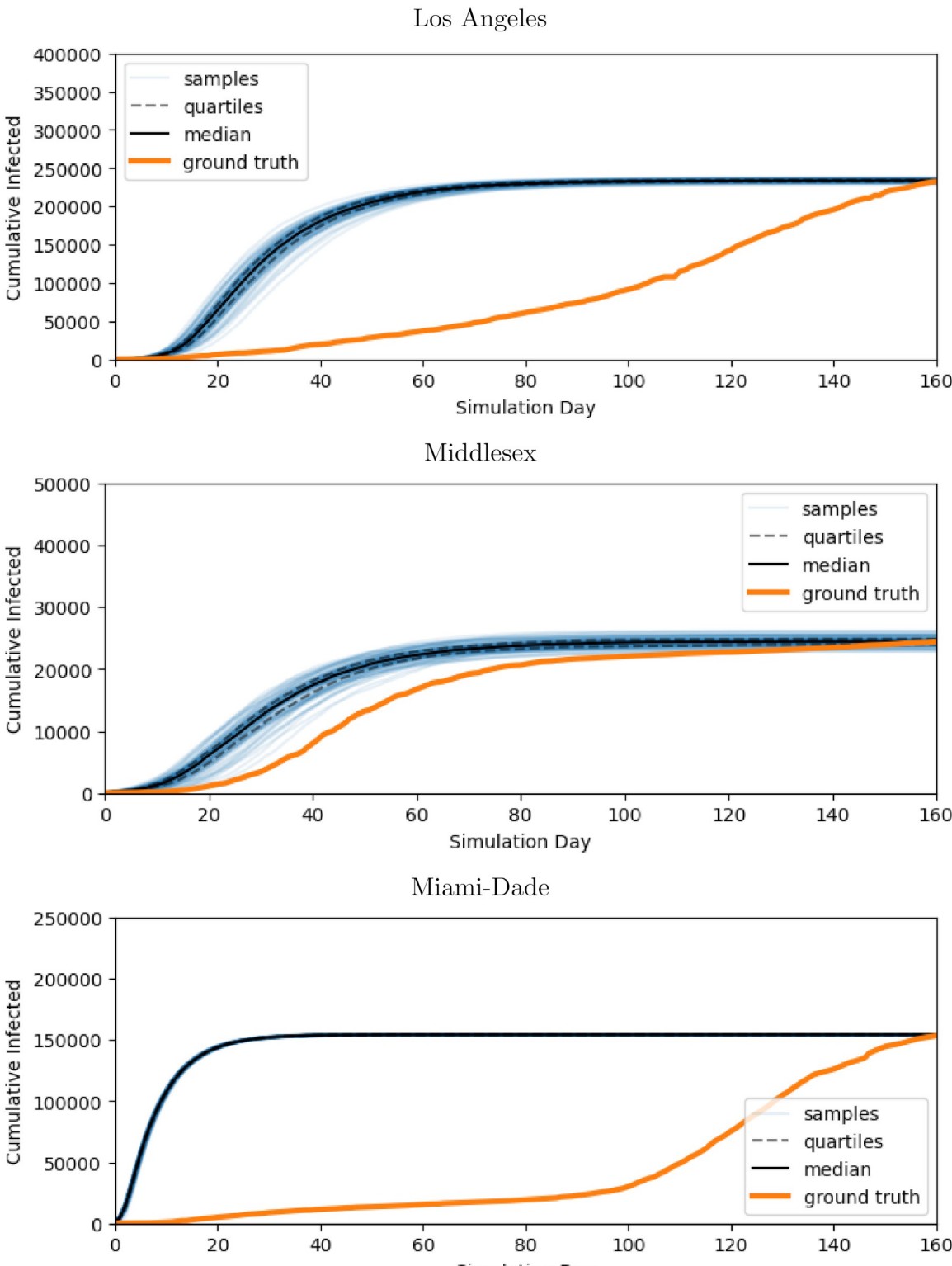

**Fig 5. $R_t$-analytic parameter inference baseline.** $R_t$-analytic derived parameters can only produce a distinctive curve shape; while this fits well for some data (such as Middlesex County above), it fits poorly much of the time.

from each chain. For IS, we include 4800 samples from the prior, with self-normalized importance weights as described in Sec. 4.3.

## 5.5 Regional variation in disease parameters

We use BBVI to fit a distribution over disease parameters for the Network-SEIR model to three diverse geographical regions in the US: Los Angeles County (CA), Middlesex County (MA), and Miami-Dade County (FL). All three counties have very different network topologies and very different historical disease trends. Nonetheless, our method is able fit all counties well and the variation in learned parameters reflects the differences in the counties well.

We fit our guide distribution over the parameters for Network-SEIR by conditioning on the cumulative infection counts in each county, as reported on the Johns Hopkins University Center for Systems Science and Engineering dashboard [39]. In Fig 6, we show network instances constructed for these three counties.

Table 3 shows the MDAE of our fits for each region and average values for $\beta_E$, the primary parameter controlling the infectiousness. Fig 7 shows the median and interquartile range of $\beta_E$ over time for all 3 regions. Fig 8 shows the posterior distribution of $\beta_E$ and $\beta_I$ at each of the 6 knots during the simulation. Note the difference in posterior variance at different time points; this may reflect the model's varying degree of confidence in the inferred parameter values. In Fig 9, we compare 100 samples from our posterior distribution to true infection counts. Fig 10 visualizes these 100 samples in terms of daily infection counts instead of cumulative counts.

We find that our network model with parameters learned through probabilistic inference fits observed data well, despite varying topological structure and disease dynamics across the different geographical regions. Our method is able to recover regional-specific disease parameters and disease progressions that reflect reported case counts. For example, in Middlesex county, where the first wave of the disease grows rapidly, our model infers higher early values for $\beta_E$ (the parameter controlling the probability of an exposed person spreading the infection). Among the three counties, we capture infections spikes occurring at different times and with different intensities. In particular, for Los Angeles county, our model is able to fit data with multiple waves of infection.

Due to the unknown initial exposure levels $\rho$, which must also be fit, we see higher variance in the early values for $\beta_t^E$. Note that inference in our probabilistic program has a degree of ambiguity; a model with lower initial exposure levels and higher initial $\beta^E$ values may give rise to similar predicted infection counts as a model with higher initial exposure levels and and lower initial infection rates. Here we hypothesize that the model may in certain cases select higher $\beta^E$ values with a lower $\rho$ in order to decrease the probability that the epidemic stops spreading due to the sub-sampling of the network topology.

## 5.6 Inferring starting communities

Since our variational distribution includes means for the proportion of initial exposure in each community $c$, we can interpret learned values for the parameters $\mu_c^\rho$ as indicating which communities were likely to have had higher initial exposure given the observed disease data. Note that this is not the same as inferring the actual precise location of the initial exposure within a region, since our cumulative global infection data is too coarse to deduce this. In other words, there are many possible initial exposure scenarios which may result in similar aggregate infection data. Rather, we can only conclude that the location of certain communities in the network topology is more consistent with observed disease dynamics. However, even having the ability to filter out potential source communities is a very useful feature in practical settings, especially when designing localized intervention strategies.

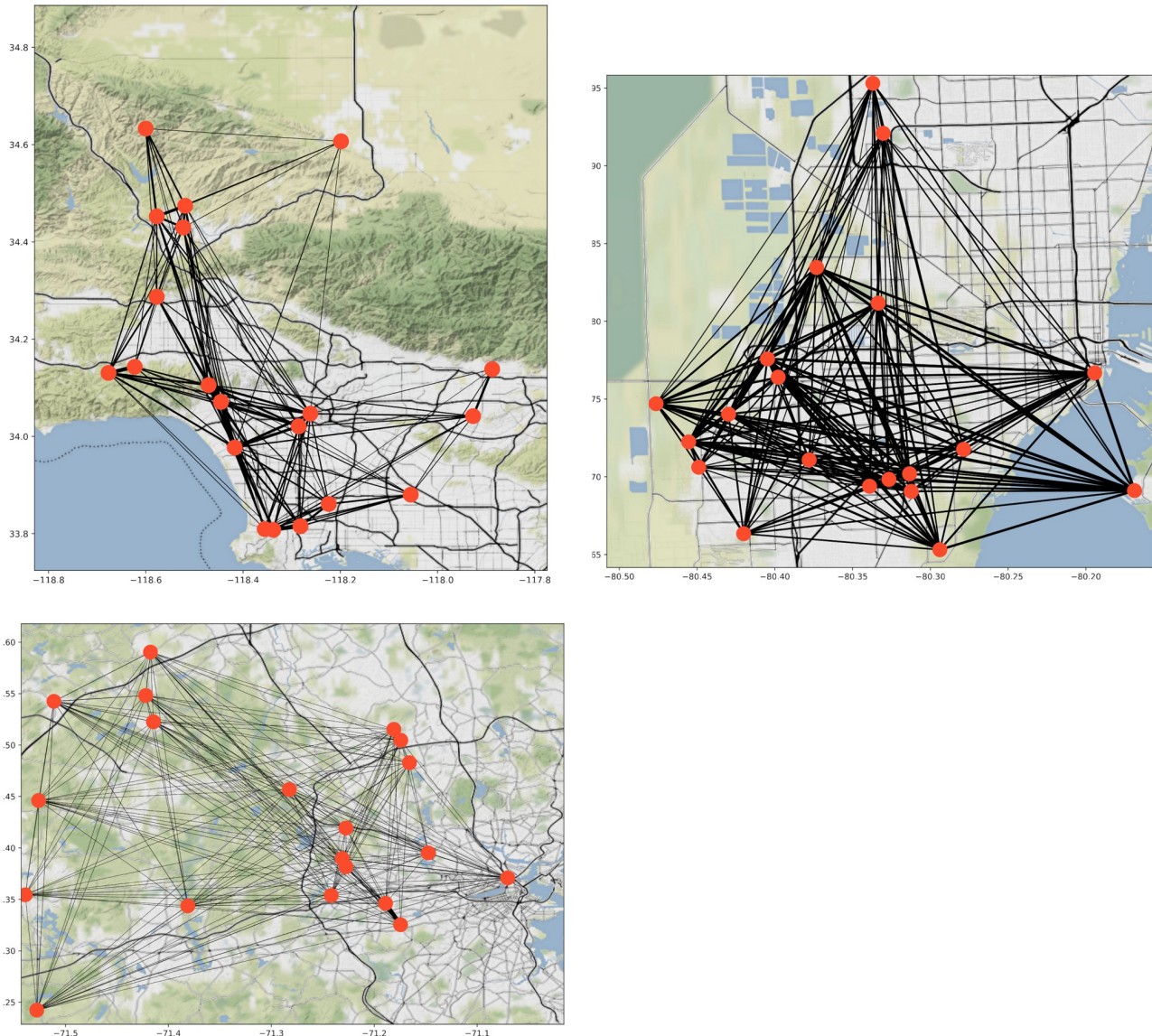

**Fig 6. Map overlay of network topologies.** Nodes from each CBG are grouped together and placed on the central coordinates for that community. Edges between CBGs represent the sum of all connected edge weights, where darker lines indicate a greater sum of edge weights. Underlying map tiles from Stamen Design under CC BY 3.0. Data by OpenStreetMap, under ODbL [53] (Top left—Los Angeles: http://maps.stamen.com/terrain-background/#10/34.0692/-118.2438. Top right—Miami-Dade: http://maps.stamen.com/terrain-background/#11/25.9046/-80.3070, Bottom—Middlesex: http://maps.stamen.com/terrain-background/#12/42.4205/-71.4415.

**Table 3. Comparison of the MDAE value and average $\beta_E$ for different regions.** Our method can fit well to different regions with different dynamics.

| Region | MDAE | Average $\beta_E$ |
|---|---|---|
| Los Angeles, CA | 0.0011 | 0.050 |
| Miami-Dade, FL | 0.0036 | 0.097 |
| Middlesex, MA | 0.0012 | 0.245 |

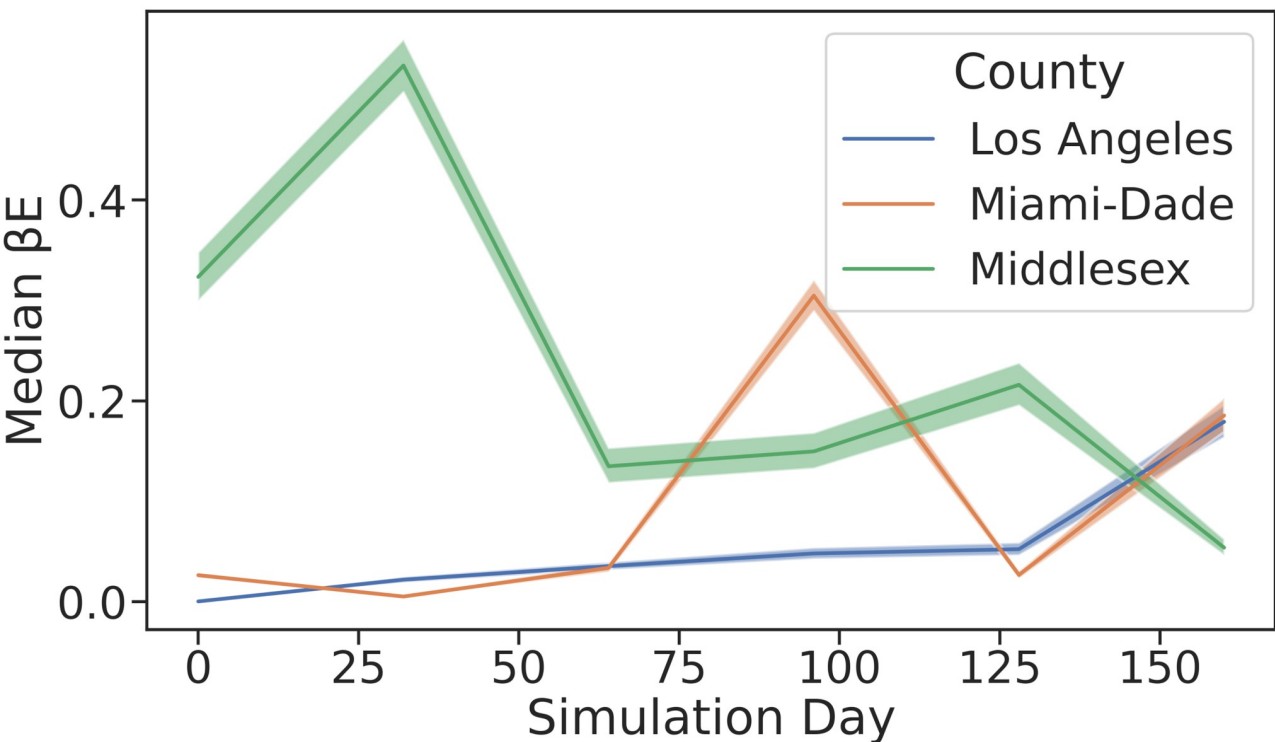

**Fig 7. Median values of $\beta_E^t$ from inferred distribution over time time.** Values are interpolated between 6 knots. Our inferred parameters vary over time to match the regional case counts; for example, in Middlesex county, our model infers high early values for $\beta_E$ due to an early spike in regional case counts.

In Fig 11, we see that for large observational noise $\nu$, the inferred parameters are closer to the uniform prior $\mu_c^\rho = .05$, whereas for small observation noise, we find a higher initial exposure in certain communities is more consistent with observed data. Varying the size of our observation noise controls the sparsity of inferred starting conditions. As our noise distribution tightens, inference moves further from our uniform prior over initial community exposure rates. For example, in the top left, in Los Angeles county, we infer higher initial exposures in communities 11 and 17.

In Fig 12, we show the posterior distribution of $\rho$ for each CBG. Recall that the total initial exposure level is fixed, so that the sampled value of $\rho_c$ only controls the fraction of initial exposure allocated to a given CBG. The posteriors shown here use the fit of lowest MDAE from each region.

## 5.7 Robustness to network size

We find that our inference procedure is flexible enough to successfully fit disease data for a range of choices in our network modeling. In particular, we show that the quality of fit is largely unaffected by the number of nodes and communities we use in our network.

Our goal is to produce a regionally-calibrated disease simulator in a practical and computationally efficient manner. Thus, we try to use the smallest networks that can still produce high-fidelity fits to regional infection statistics. We vary network size by selecting the top subset of most highly connected CBGs after constructing the block matrix $P$ in our DCSBM. This preserves as much disease transmission as possible between communities, while also allowing us

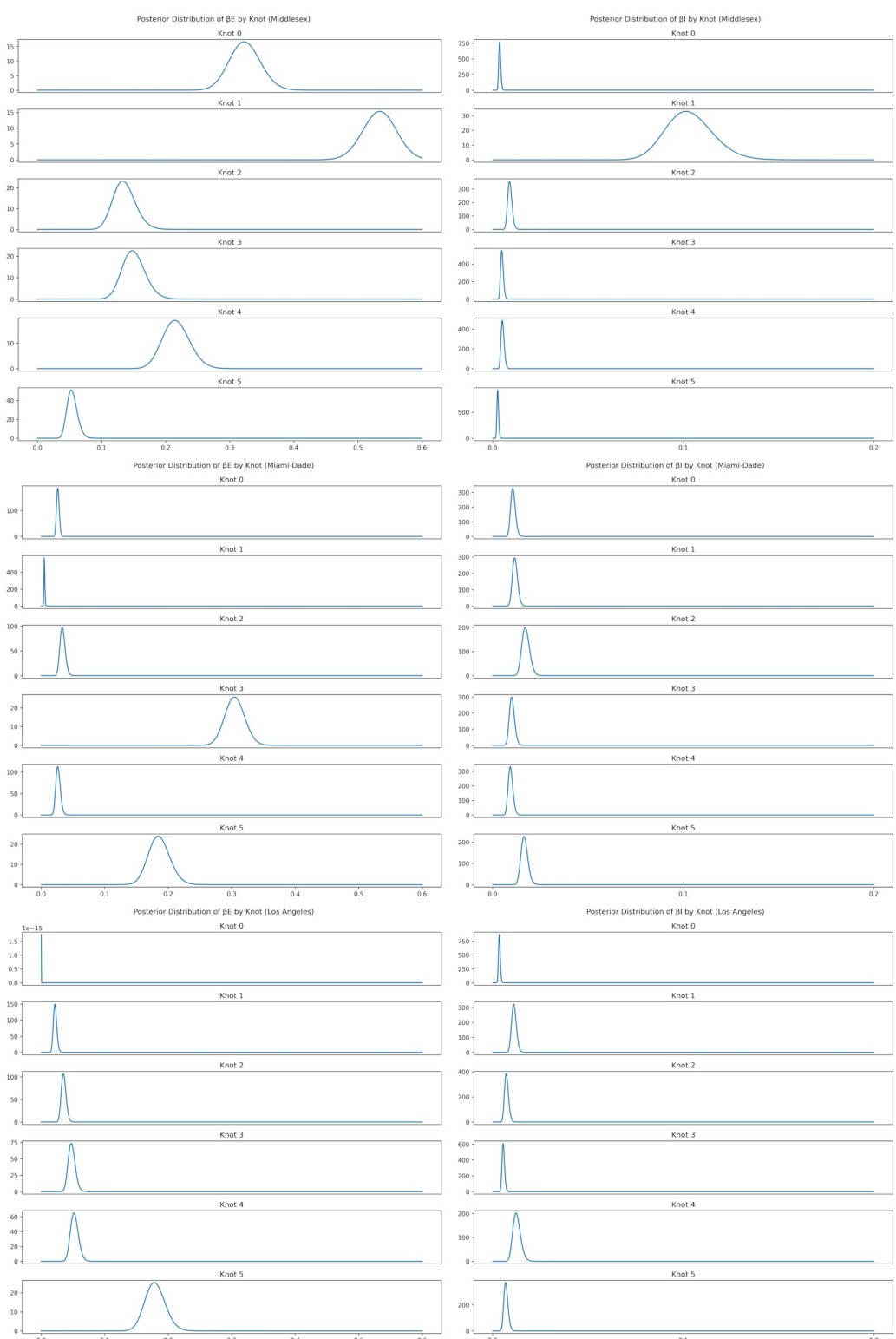

**Fig 8. Posterior distribution of $\beta_E^{t_n}$ and $\beta_I^{t_n}$ at each change point during simulation.**

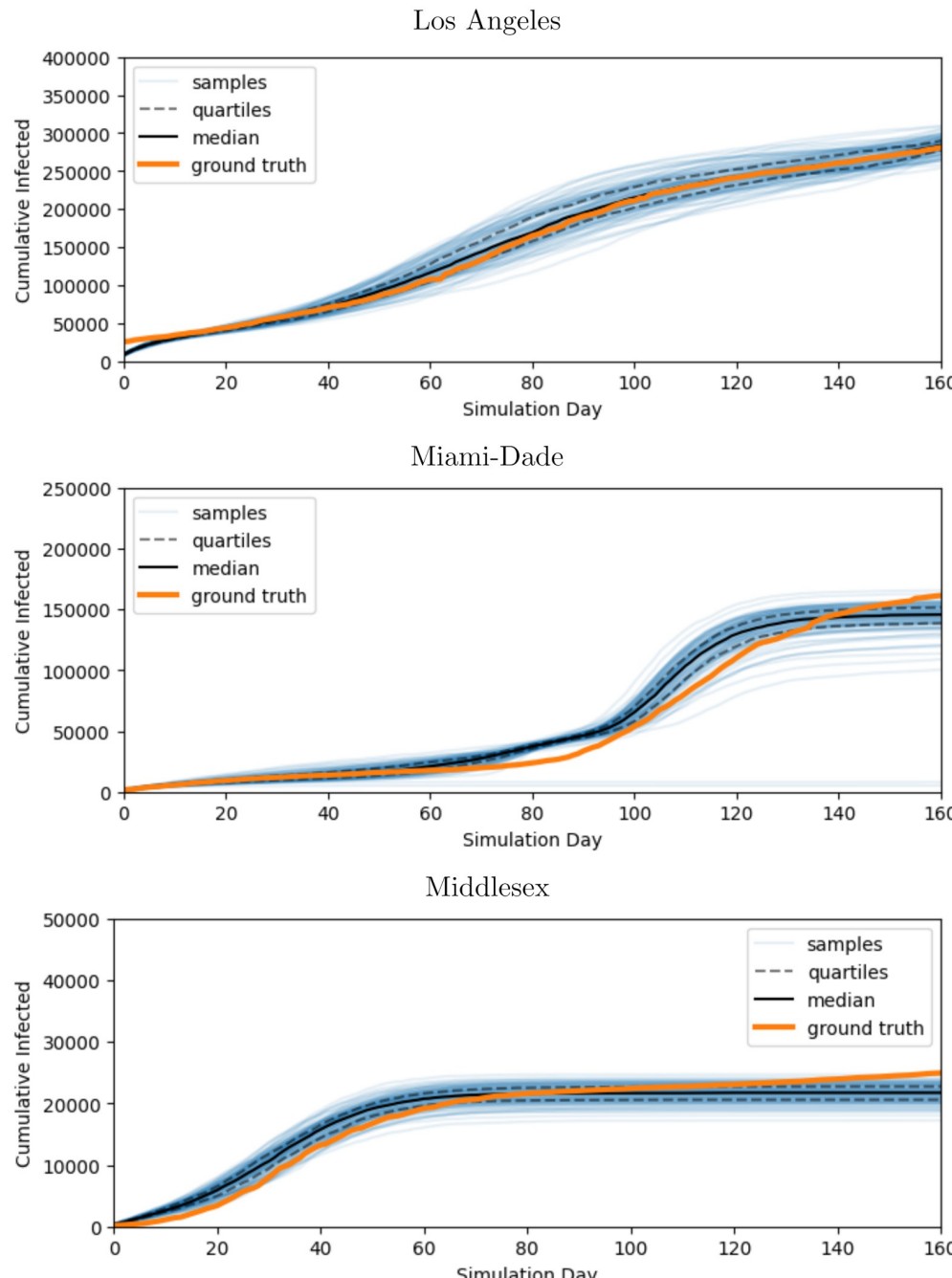

**Fig 9. Cumulative infection trajectories sampled from fitted model.** Our method is capable of obtaining distributions of disease parameters which reproduce true data closely over a variety of regions. The orange line represents true cumulative infections counts for 160 days starting from 7 days before the first day in which infections counts accounted for 0.5% of the population: May 3, 2020 for Los Angeles, March 29, 2020 for Miami-Dade, and March 15, 2020 for Middlesex. The blue lines represent 100 simulations of $f_{\mathrm{NSEIR}}$ using disease parameters sampled from our fit variational distribution. The black lines represent quartiles for these 100 samples.

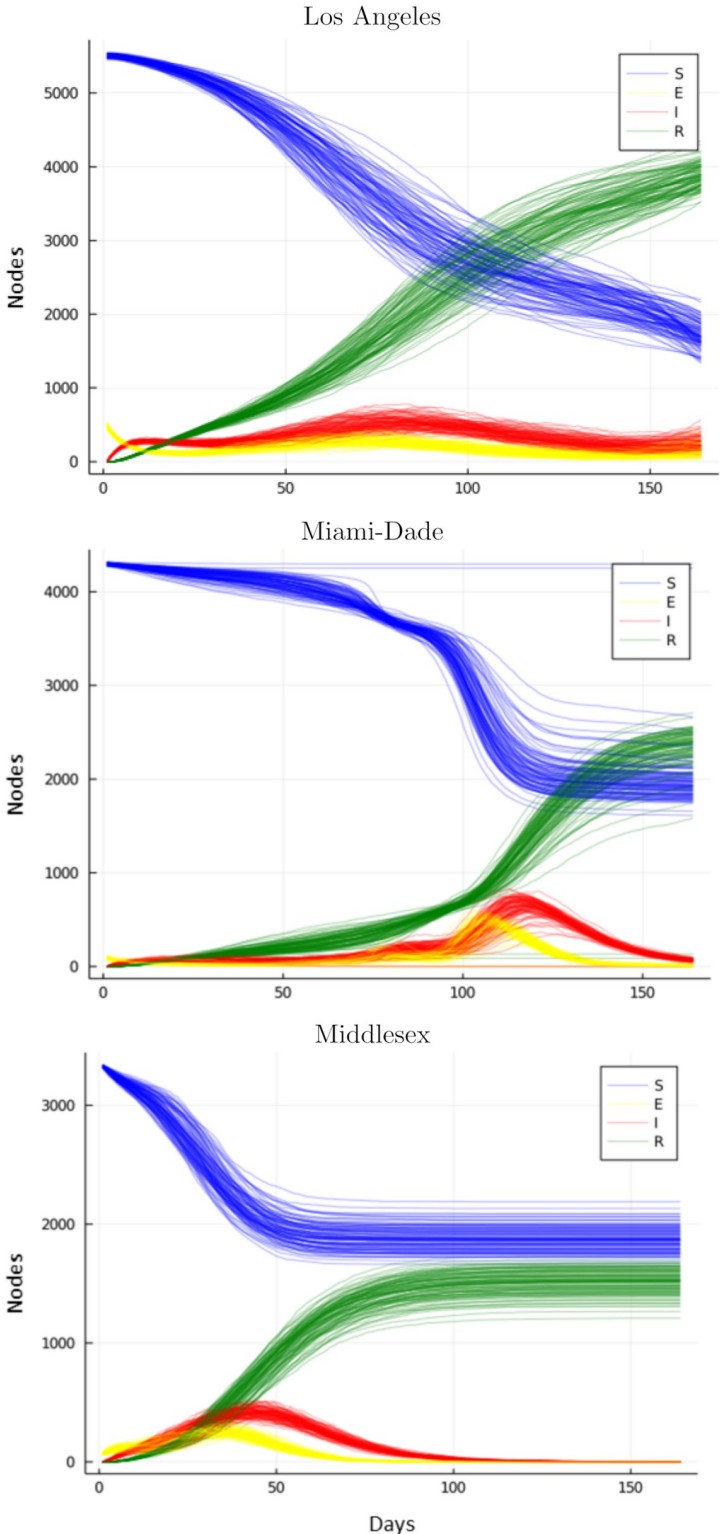

**Fig 10. SEIR curves produced by fitted model.** Our method is capable of fitting different disease dynamics in different regions including infections with multiple waves and different rates of infectivity over time. We plot the total Susceptible, Exposed, Infected, and Removed (SEIR) counts over 160 days from 100 simulations for our fit posterior distribution.

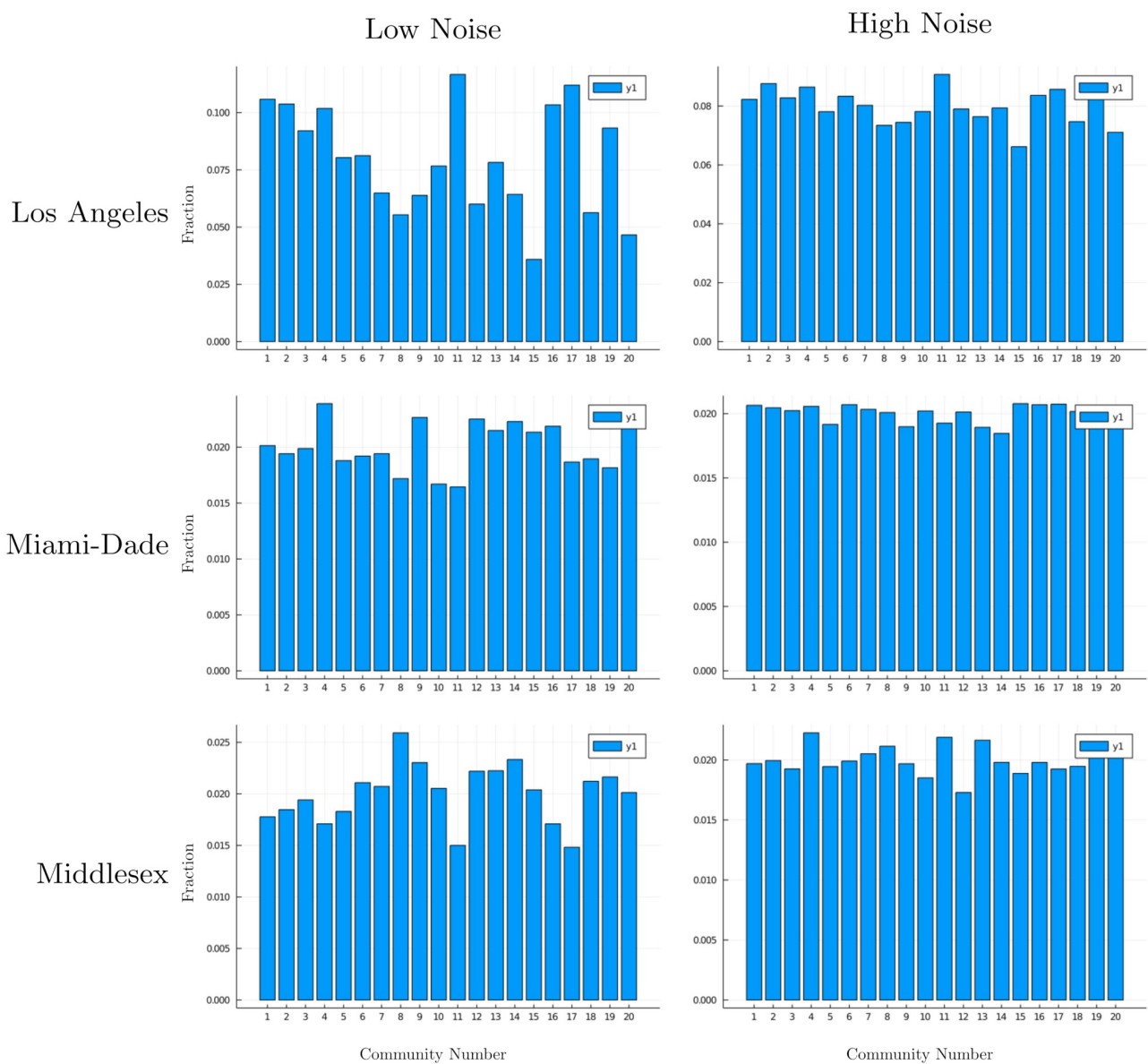

**Fig 11. Varying observation noise level controls sparsity of inferred starting conditions.** As noise distribution tightens, inference moves further from our uniform prior on initial community exposure rates. Low noise corresponds to $\nu = 0.00025$, and high noise to $\nu = 0.0005$. The network topology of each county is modeled using 20 communities which correspond to actual geographic areas. We plot $\mu_c^\rho$ for $1 \le c \le 20$. In the left plots, we use $\nu = 0.00025$, a tighter observational noise than the right plots where $\nu = 0.0005$.

to trade-off fidelity of our network model with speed and efficiency of our overall disease simulator.

In Fig 13 and Table 4, we compare the disease models fits on graphs of varying size with either 5, 10, 15, 20, 30, or 40 CBGs. We see that applying the same prior parameters on these different size graphs produces substantially different prior behavior due to the effect of the different topologies on disease spread. In particular, since we scale the infection data to the down-sampled graph before parameter fitting, the infection must spread more slowly from node to node to achieve the same scaled trajectory. For highly downsampled graphs, we thus observe smaller $\beta_E$ values. Nonetheless, we observe that our inference can successfully find parameter

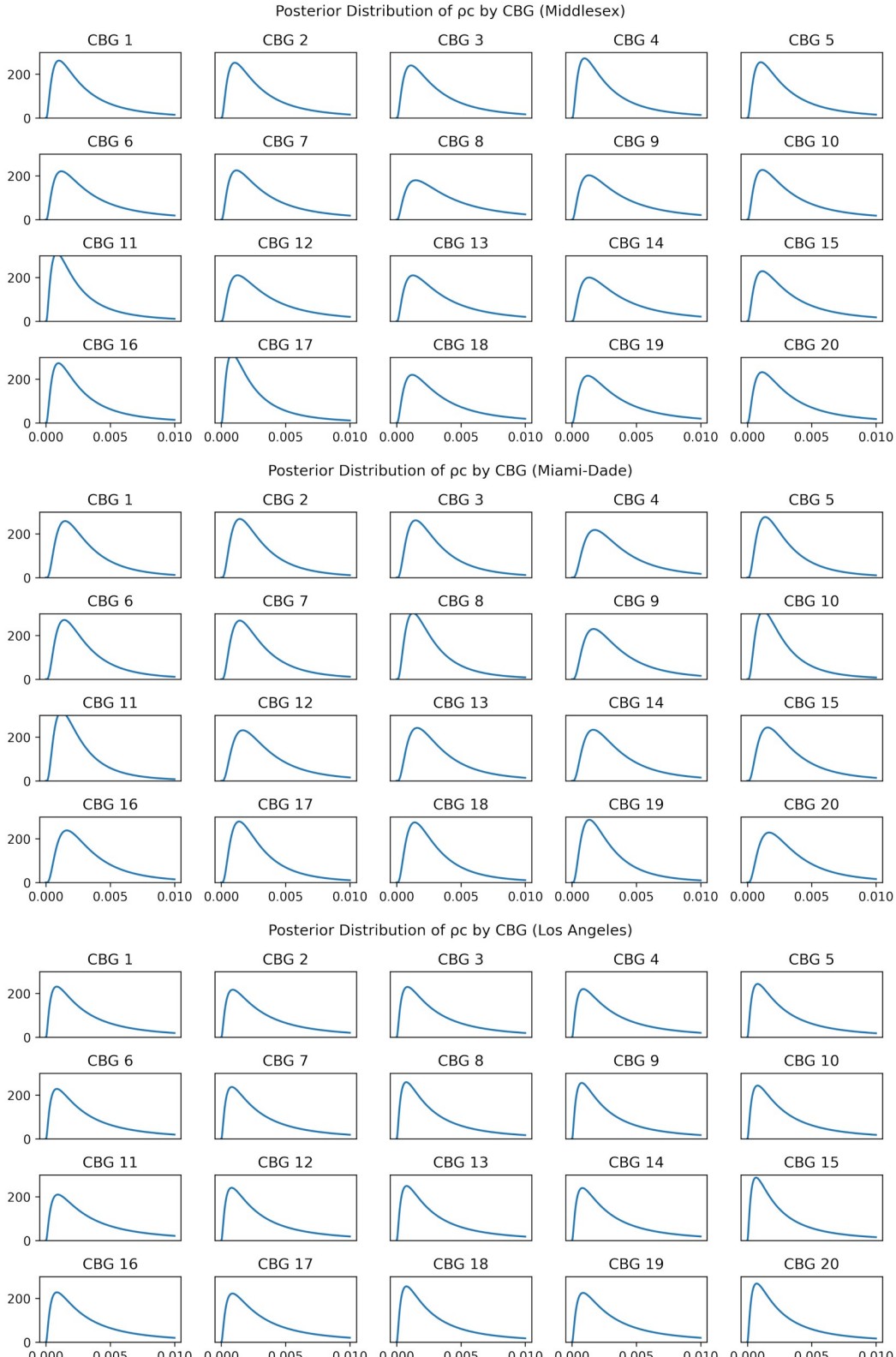

**Fig 12. Posterior distribution of $\rho_c$ for communities and $\beta_i^t$ at each change point during simulation.** Note that CBGs have no correspondence across counties.

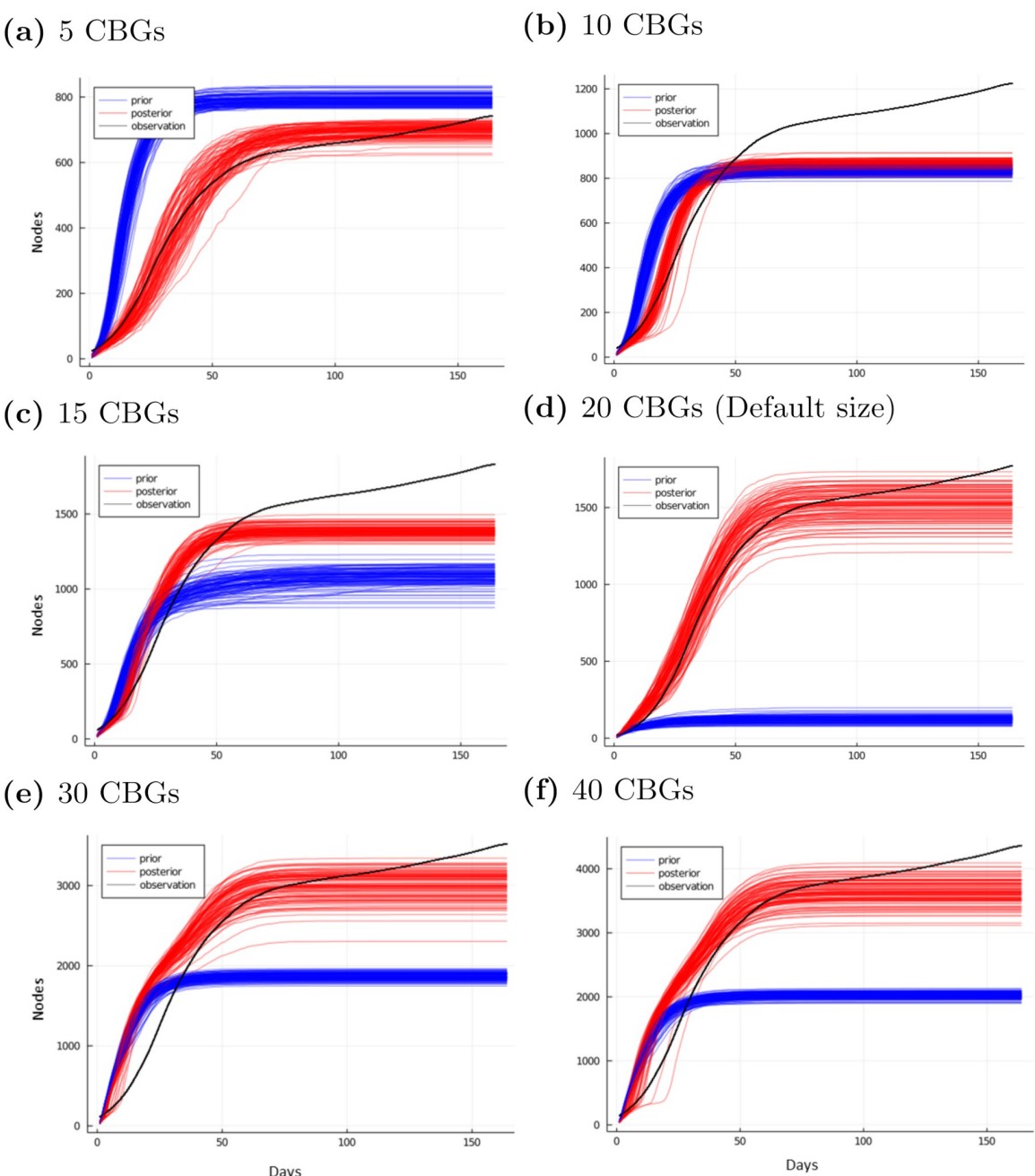

**Fig 13. Prior and posterior disease trajectories with varying network size.** Applying the same prior parameters on graphs subsampled to a different initial set of CBGs produces substantially different prior behavior for the disease simulator (blue). Our inference converges to a consistent behavior (red) that is close to the observed data (black).

values that fit the observed data well, and thus the MDAE does not vary significantly for different sizes of subsampled graph. Note that we use 20 CBGs for all other experiments.

In Table 4 and Fig 14, we compare the inferred values of $\beta^E$ for each graph. Since the network topologies vary, we should not necessarily expect to find consistent values for different numbers of CBGs. Nonetheless, the parameters are fairly consistent during the middle and late

**Table 4. The MDAE does not vary significantly for different sizes of subsampled graphs.** The inference procedure found values of $\beta^E$ which fit the observed data well in each case.

| Num. CBG | MDAE | Average $\beta_E$ |
|---|---|---|
| 5 | 0.00077 | 0.055 |
| 10 | 0.00239 | 0.072 |
| 15 | 0.00196 | 0.111 |
| 20 | 0.00118 | 0.245 |
| 30 | 0.00159 | 0.191 |
| 40 | 0.00153 | 0.177 |

time-range. There is more variance in the inferred early values of $\beta^E$, likely due to the model compensating for underestimates of initial exposure.

## 5.8 Sensitivity to time-varying network model

We find that our model still fits well if we use time-varying edges weights to reflect changing mobility patterns in the network. In this case, the $\beta^E$ and $\beta^I$ parameters only reflect changing infectivity patterns.

Transmission dynamics change as a result of at least two sources of variation. First, changes in mobility and interaction frequency may occur due to quarantines and business closures. Second, transmission probability during each contact may change, due to the use of personal protective equipment, hygiene practices, seasonal effects, and disease mutation. These

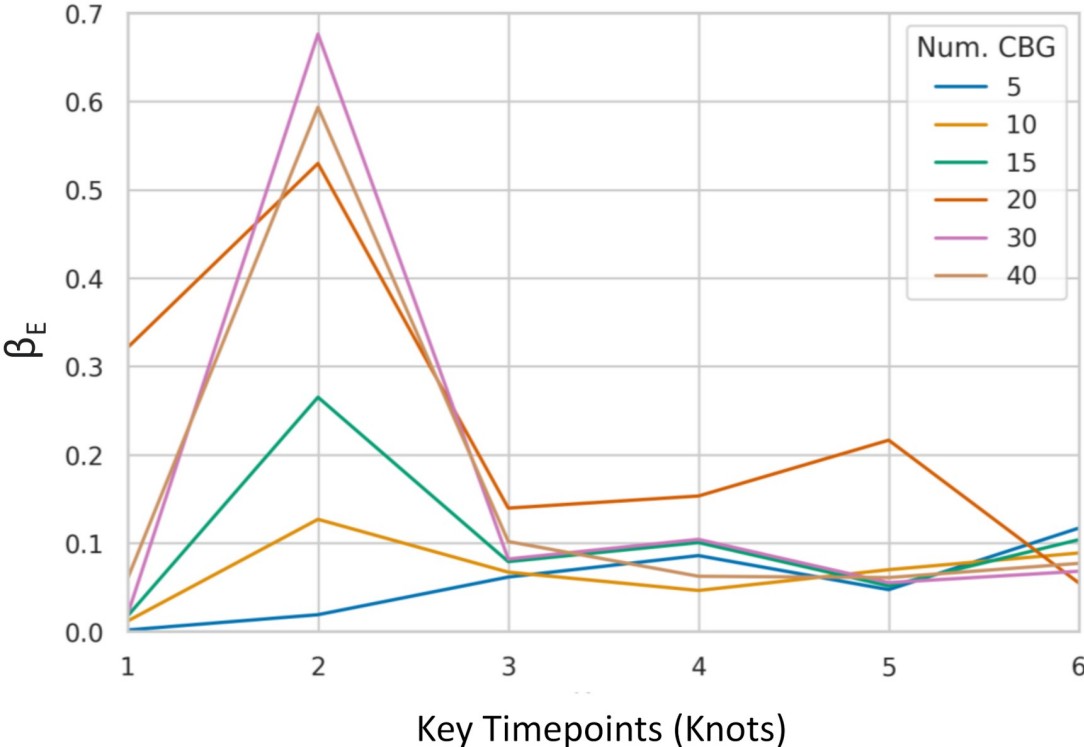

**Fig 14. Posterior mean parameter values with varying network size.** We vary the size of our simulated network by varying the number of Census Block Groups (CBG) used during construction. We observe that the inferred disease parameters follow a similar trend even across large differences in network size.

non-pharmaceutical interventions may have a strong effect on the trajectory of the pandemic. In our model as described in Sec. 4.3, we account for both of these sources of variation by allowing our disease transmission parameters $\beta^E$ and $\beta^I$ to vary over time. Alternatively, we can account for variation in mobility in the network itself, by varying the network across time. This leaves the parameters $\beta^E$ and $\beta^I$ to model the transmission probability during each contact.

In this experiment, we evaluate whether using time-varying network connectivity can further improve the quality of fit. To do so, we construct time-varying networks as follows. Similar to Sec. 4.2.1, each estimate of network structure is formed from a week's worth of geo-location data. At the beginning of each week of our disease simulator, we allow the network structure to vary. We begin with a network estimated from the first week of data as before. For subsequent weeks, we separately estimate a network instance for each week of data, containing the same CBGs and the same number of nodes in each CBG. All nodes are assigned by ID to the same households, to maintain an identity for each node, and other edges (both connectivity and weights) are allowed to vary freely.

Note that procedure allows for the possibility that nodes will change rank order (i.e. that the highest degree node for one week may not be highest in another week), since constraining nodes to keep the same rank order may be unrealistic. Note also that we continue to allow disease parameters to vary over time; in this way, we expect that the network modeling can capture mobility-related effects, while the time-varying disease parameters can capture the behavioral and other effects mentioned above.

In Fig 15, we find that the model successfully learns time varying parameters that are relatively consistent with the observed data, achieving an MDAE of 0.00260. We hypothesize that

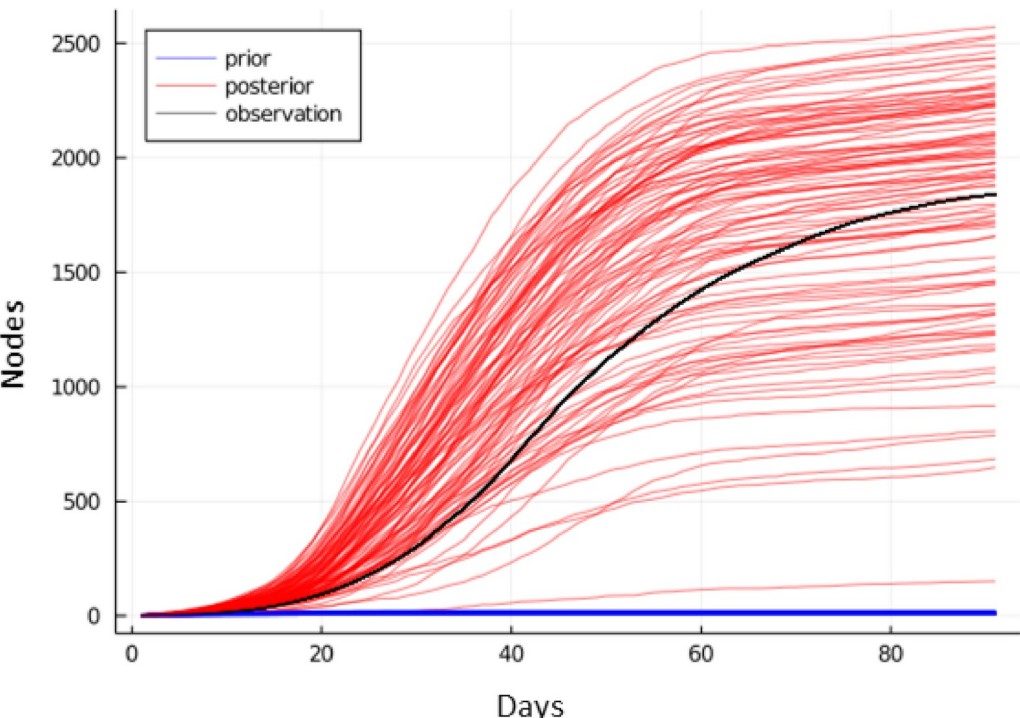

**Fig 15. Inferred cumulative infection statistics and SEIR curves for network modeled with time-varying edge weights.** Our model still finds parameters that approximately match the data, even when the network topology changes over time. Note that the model also compensates for the poor performance of the prior parameters.

the relatively noisy procedure used for node re-assignment contributes additional error to our model.

## 5.9 Using death data for model fitting and evaluation

The model and metric may be simply extended to use cumulative death counts in addition to cumulative infection counts. Due to issues with infection testing in the early days of the pandemic, infection data was prone to undercounting and delay. While undercounting and delay may also be an issue for deaths in cases when the correct cause of death is not ascertained, death statistics are often more reliable that infection counts. While death counts may be more reliable, they are also difficult to use in a small-scale model, since the scaled number of deaths may be less than a single node. This results in a potential resolution issue during inference.

We modify Algorithm 4 to also return the size of the removed compartment $\{R_j\}_{j=1}^{T}$ at all times. The removed compartment includes both recovered and deceased individuals and so we estimate daily death counts $d_t = \lambda_D R_t$ where $\lambda_D$ is the mortality rate in the given county for the given time period, derived from case data [39]. While it would be more accurate to add a separate death compartment to Network-SEIR with time-varying mortality rate, our goal here is to show that death data may be used to condition our existing disease simulator with minimal modifications.

We then modify the likelihood of our model (Eq 25) by comparing both modified model outputs $s_{1:T}$ and $d_{1:T}$ to the case counts $x_{1:T}$ and death counts $x_{1:T}^d$ for a particular county in a particular time range

$$x_t \sim \mathcal{N}(r\,s_t, \ r\,\sigma^x(G, v, t)), \qquad \sigma^x(G, v, t) = v\sqrt{t}|\mathcal{V}|, \tag{31}$$

$$x_t^d \sim \mathcal{N}(r\,d_t, \ r\,\sigma^x(G, v, t)). \tag{32}$$

Thus our inference procedure will find parameters whose resulting simulator dynamics are in agreement with both observed infections and observed deaths.

To evaluate the effectiveness of this modified model, we evaluate the model using MDAE (Eq 27) to compare model output infections $s_{1:T}$ to observed case counts $x_{1:T}$ and model output deaths $d_{1:T}$ to observed death counts $x_{1:T}^d$.

In Fig 16 and Table 5, we find that the model is able to assimilate the new input signal coming from the death data and infer model parameters which result in good fits for both observed death and infection counts. Note that the MDAE metric is scaled by county population, so we naturally observe a lower value when measuring prediction error for death counts.

## 5.10 Simulator noise and posterior variance

One of the challenges of fitting the parameters of a stochastic simulator is that the intrinsic randomness of the simulator sets a limit on the level of precision of the learned parameters. In Fig 1, we show examples of running the stochastic disease simulator using the mean parameters $\bar{\theta} = \mathbb{E}_{\theta \sim q_\phi}[\theta]$, or using samples from the posterior distribution $\theta \sim q_\phi$. In both cases, we sample trajectories $s$ from the simulator. This allows us to visualize the difference between the level of randomness inherent in the simulator and the variance of the posterior distribution over model parameters. Note that these results are produced using the same runs discussed in Section 5.9; a similar pattern occurs for results in all of our experiments.

In Fig 17, we show the average ELBO value over time for the same runs, to show that the fitting procedure has converged well in these experiments. At this point in the model optimization, it is likely the case that the untraced randomness of the simulator model has become the

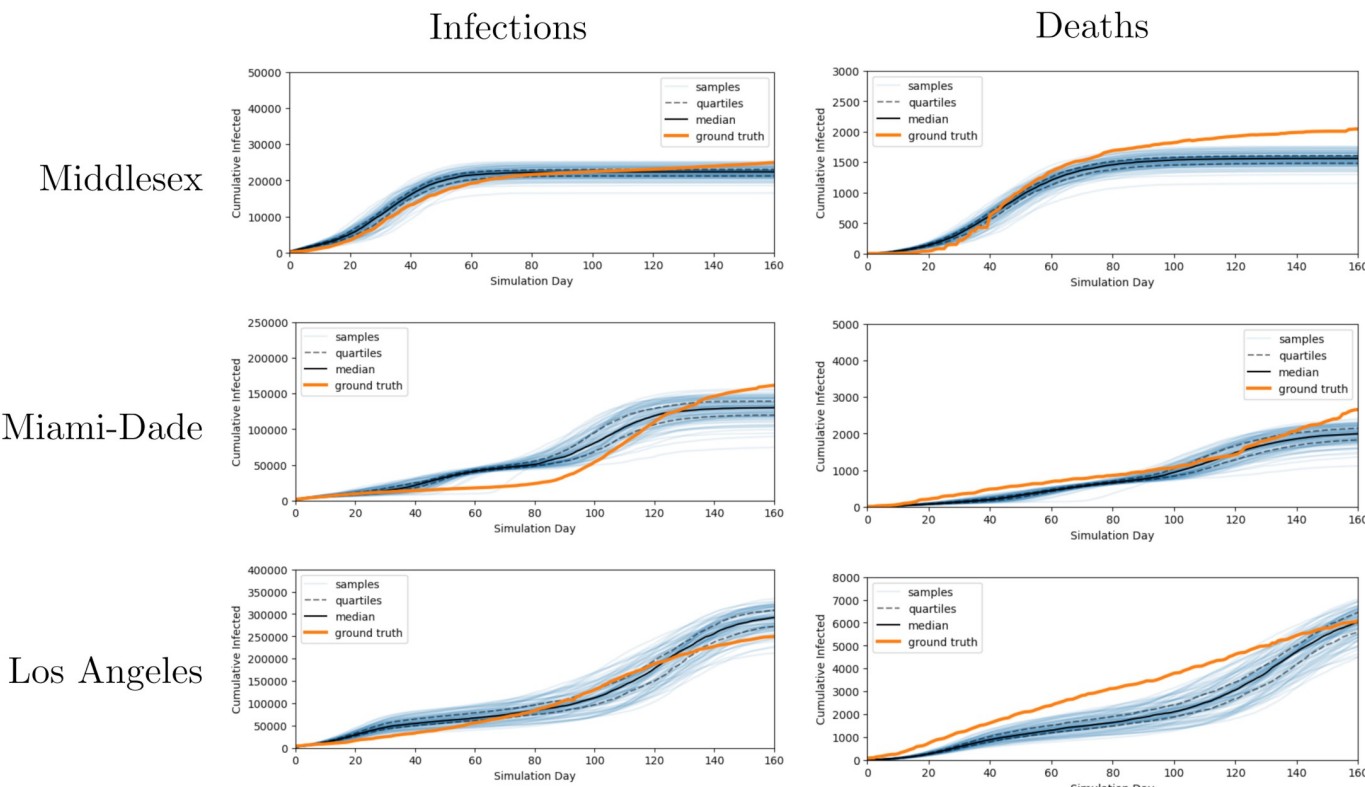

**Fig 16. Cumulative infection trajectories sampled from model fit to infection and death data.** We see that posterior samples of cumulative infection and death counts from several counties are generally in good agreement with the data when we use both infections and deaths as input observations to our model.

limiting factor in the quality of the estimated ELBO gradient, and thus we expect that the variance of the posterior likely can not be decreased much further. In other words, while sampling parameters to estimate the gradient at the next iteration, even if the model samples parameter values corresponding to the maximum marginal likelihood estimate, the significant randomness of the simulator means that these may not achieve a much higher likelihood value than a slightly different set of parameter values. Overcoming this level of inherent noise in the simulator would likely require greatly increasing the sample budget.

## 6 Discussion

The results in this paper demonstrate that probabilistic programming methods are a viable tool for parameter inference in epidemiological simulations that have previously been implemented in general purpose languages. The likelihood weighting, variational inference, and Metropolis-Hastings methods that we discussed are among the simplest and most widely used

**Table 5. MDAE for model conditioned on infection and death statistics.** Note that MDAE is scaled relative to the population of a given county; since death counts are smaller, a model with the same relative quality achieves lower absolute error.

| County | Infections MDAE | Deaths MDAE |
|---|---|---|
| Middlesex | 0.0011 | 0.0002 |
| Miami-Dade | 0.0065 | 0.00008 |
| Los Angeles | 0.0022 | 0.00009 |

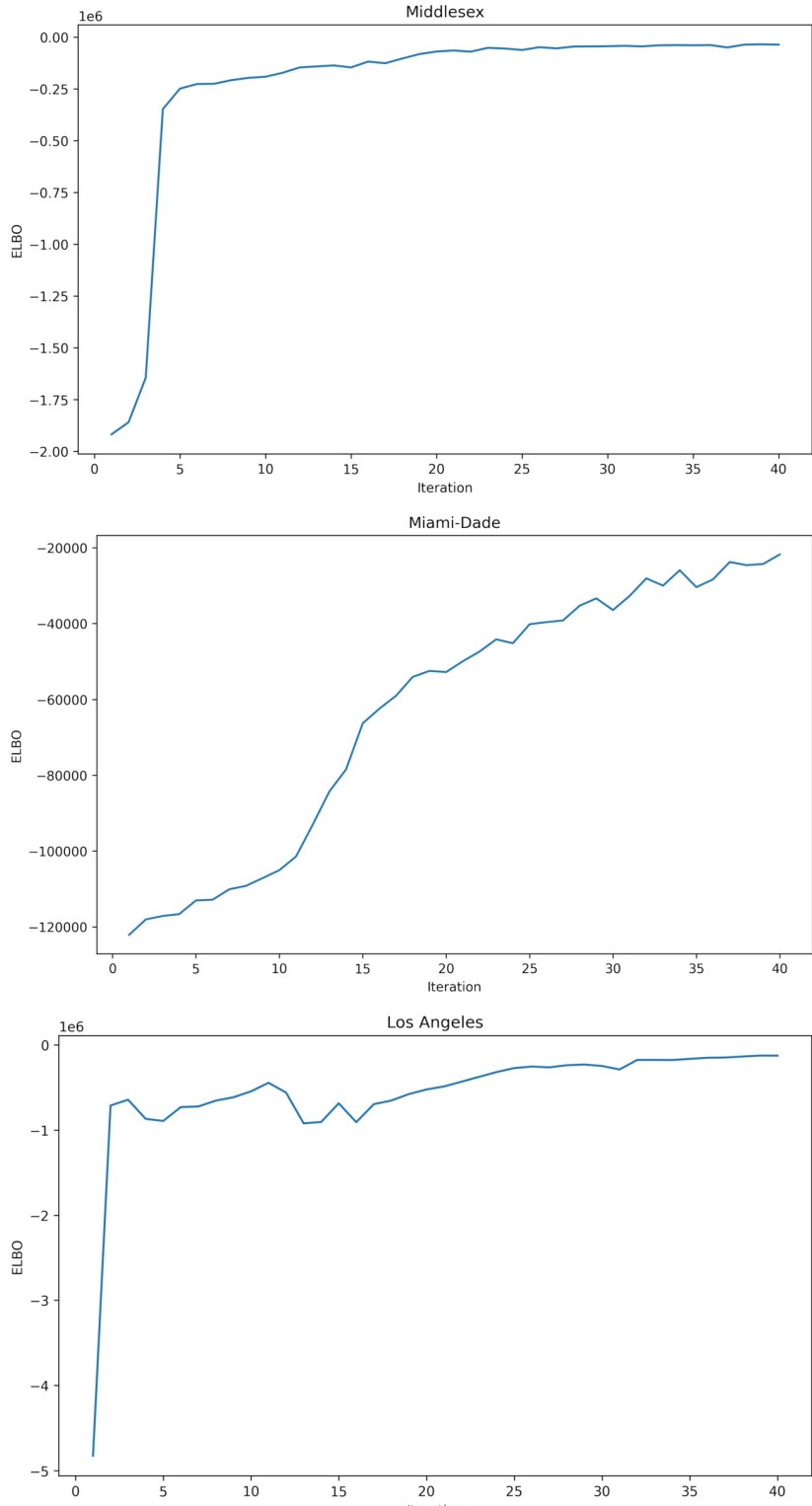

**Fig 17. Convergence curves, showing the ELBO at each iteration of optimization.** We see that parameter inference has converged within the allocated computation budget.

in the probabilistic programming literature. For epidemiologists looking to apply these inference methods to their models, there exist mature system implementations for models in Python [15], Julia [18, 42], Clojure [43], and Javascript [44]. Moreover, systems like PyProb [45] make it possible to apply inference methods to programs written in most languages by way of a cross-language distribution library. This means that researchers designing disease simulators have a much broader array of inference methods at their disposal than the standard likelihood-weighting and ABC rejection algorithms that are commonly used to perform inference in large-scale granular disease simulations. Our experiments suggest that investing the time to apply these methods is likely worthwhile; even comparatively simple methods can substantially improve the quality of fit in such models relative to baseline inference techniques.

As we mentioned at the start of this paper, the network-based disease model that we employ has a number of limitations from an epidemiological perspective. Our simulator is relatively granular; it implements an agent-based simulation defined in terms of region-specific mobility networks and is conditioned on region-specific case count. However, we make simplifying assumptions, such as considering a subsampled population of 2000 to 15000 nodes. This means we must rescale our model output to compare with reported case counts, and inferred parameter values will depend on the degree of subsampling. Moreover, simulations are conditioned on aggregate county case counts, which means that the problem of inferring initial conditions in the simulation is highly under-constrained. We would however expect our approach to perform better in this regard with more granular case counts (i.e. counts at CBG level). Finally, using time-dependent transmission parameters means that adapting this model to perform forecasting would require additional smoothing assumptions. A promising approach in this context would be to replace the linear interpolation with a (variational) Gaussian process prior [46, 47].

There are also technical limitations to the inference methods that we employ. We simplify the design of our variational distribution by factoring it into a product over different parameters. This approximation is commonly used in many probabilistic programming strategies, because it lowers the dimension of the parameters, making inference more tractable, while often being sufficiently descriptive to achieve good model fits. However, this use of a fully-factorized approximation is known to produce a posterior distribution which is known to under-approximate the posterior variance. This means that uncertainty estimates for parameters are not necessarily reliable. Note that many simple inference baselines such as likelihood-weighted importance sampling may suffer from an analogous problems. Using LW, a small subset of samples will often receive the majority of the weight, resulting in a very tightly peaked posterior with a poor estimate of variance. A second limitation is that we employ simple score-function estimators for optimizing variational parameters using stochastic gradient descent [16, 17]. These are generally applicable to models that employ non-differentiable functions, but are known to yield high-variance gradient estimates. There is a large body work for probabilistic program inference that improves upon standard score-function estimators using reparameterization (which requires differentiable models) [48], neural proposals [45, 49], or by combining variational inference and importance sampling [50, 51, 52]. We leave application of such methods to future work.

## Supporting information

**S1 Appendix. Derivation of blackbox variational inference.** We restate the derivation of the technique of blackbox variational inference, including the ELBO objective and the score function ELBO gradient.
(PDF)

**S2 Appendix. Baseline methods.** We explain the details of the Analytic $R_t$-matched parameter estimation and Certainty-Equivalent Expectation Maximization baseline methods.
(PDF)

**S3 Appendix. Additional experimental details and results.** We explain our hyperparameter selection. We describe how the initial exposed fraction was handled for the baseline $R_t$ analytic and CE-EM methods. Finally, we show results of the CE-EM baseline method.
(PDF)

**S1 Fig. Daily SEIR and cumulative infection counts simulated using SEIR equations using parameters estimated by CE-EM.** This model is only capable of outputting a disease history corresponding to a single wave of infection. In Miami-Dade, this allows for a reasonable approximation of the regional case counts, whereas for Middlesex and Los Angeles the fit is much worse.
(TIFF)

## Acknowledgments

Computation was performed using the Northeastern Discovery and MIT Supercloud clusters. We thank Marco Cusumano-Towner for help with the Gen.jl package. We are also grateful to Tina Eliassi-Rad, Mykel Kochenderfer, Kunal Menda, Leo Torres, Ross Alexander, Shushman Choudhury, and Chris Rackauckas who worked on other aspects of Network-SEIR and have contributed vastly to improving and understanding the model.

**Distribution statement A**. Approved for public release. Distribution is unlimited. This material is based upon work supported by the Under Secretary of Defense for Research and Engineering under Air Force Contract No. FA8702-15-D-0001. Any opinions, findings, conclusions or recommendations expressed in this material are those of the author(s) and do not necessarily reflect the views of the Under Secretary of Defense for Research and Engineering. Copyright 2022 Massachusetts Institute of Technology. Delivered to the U.S. Government with Unlimited Rights, as defined in DFARS Part 252.227-7013 or 7014 (Feb 2014). Notwithstanding any copyright notice, U.S. Government rights in this work are defined by DFARS 252.227-7013 or DFARS 252.227-7014 as detailed above. Use of this work other than as specifically authorized by the U.S. Government may violate any copyrights that exist in this work.

## Author Contributions

**Conceptualization:** Rajmonda Caceres, Jan-Willem van de Meent.

**Data curation:** Lucas Laird, Christian van der Loo, Neela Kaushik.

**Formal analysis:** Niklas Smedemark-Margulies, Heiko Zimmermann, Rajmonda Caceres, Jan-Willem van de Meent.

**Funding acquisition:** Rajmonda Caceres, Jan-Willem van de Meent.

**Investigation:** Niklas Smedemark-Margulies, Robin Walters, Heiko Zimmermann.

**Methodology:** Niklas Smedemark-Margulies, Robin Walters, Heiko Zimmermann, Jan-Willem van de Meent.

**Project administration:** Rajmonda Caceres, Jan-Willem van de Meent.

**Resources:** Rajmonda Caceres, Jan-Willem van de Meent.

**Software:** Niklas Smedemark-Margulies, Robin Walters, Heiko Zimmermann, Lucas Laird, Neela Kaushik.

**Supervision:** Rajmonda Caceres, Jan-Willem van de Meent.

**Validation:** Niklas Smedemark-Margulies, Robin Walters, Heiko Zimmermann.

**Visualization:** Niklas Smedemark-Margulies, Robin Walters, Heiko Zimmermann, Christian van der Loo.

**Writing – original draft:** Niklas Smedemark-Margulies, Robin Walters, Heiko Zimmermann, Jan-Willem van de Meent.

**Writing – review & editing:** Niklas Smedemark-Margulies, Robin Walters, Heiko Zimmermann, Lucas Laird, Neela Kaushik, Rajmonda Caceres, Jan-Willem van de Meent.

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
