## [Decision Letter · Decision Letter 0]

12 May 2022

Dear Dr. Walters,

Thank you very much for submitting your manuscript "Probabilistic Program Inference in Network-based Epidemiological Simulations" for consideration at PLOS Computational Biology.

As with all papers reviewed by the journal, your manuscript was reviewed by members of the editorial board and by several independent reviewers. In light of the reviews (below this email), we would like to invite the resubmission of a significantly-revised version that takes into account the reviewers' comments.

We cannot make any decision about publication until we have seen the revised manuscript and your response to the reviewers' comments. Your revised manuscript is also likely to be sent to reviewers for further evaluation.

Sincerely,

Tom Britton

Deputy Editor

PLOS Computational Biology

Reviewer's Responses to Questions

**Comments to the Authors:**

Reviewer #1: Please see my report attached.

Reviewer #2: See attachment

**Have the authors made all data and (if applicable) computational code underlying the findings in their manuscript fully available?**

Reviewer #1: Yes

Reviewer #2: **No: **All code and data will be publicly available at time of publication.

PLOS authors have the option to publish the peer review history of their article (what does this mean?). If published, this will include your full peer review and any attached files.

Reviewer #1: No

Reviewer #2: No
---

## [Decision Letter · Decision Letter 1]

21 Sep 2022

Dear Dr. Walters,

We are pleased to inform you that your manuscript 'Probabilistic Program Inference in Network-based Epidemiological Simulations' has been provisionally accepted for publication in PLOS Computational Biology.

Best regards,

Tom Britton

Section Editor

PLOS Computational Biology

Reviewer's Responses to Questions

**Comments to the Authors:**

Reviewer #1: In the original submission, the authors developed a Network-SEIR Model, which comprises two components: (i) a network topology model and (ii) an agent-based compartmental model. The authors also used probabilistic program inference methods to approximate the distribution over disease transmission parameters. In this revision, the authors did a great job of incorporating the suggestions of the reviewers and improving

the original article by emphasizing more on the study of the sensitivity to the time-varying network model and adding more discussion about the motivation, advantages and limitations of the proposed method.

Reviewer #2: The authors have addressed all my prior concerns and I do not have further comments

**Have the authors made all data and (if applicable) computational code underlying the findings in their manuscript fully available?**

Reviewer #1: Yes

Reviewer #2: None

PLOS authors have the option to publish the peer review history of their article (what does this mean?). If published, this will include your full peer review and any attached files.

Reviewer #1: No

Reviewer #2: No

---

## [Editor Report · Acceptance letter]

25 Oct 2022

PCOMPBIOL-D-21-02223R1 

Probabilistic Program Inference in Network-based Epidemiological Simulations

Dear Dr Walters,

I am pleased to inform you that your manuscript has been formally accepted for publication in PLOS Computational Biology. Your manuscript is now with our production department and you will be notified of the publication date in due course.

With kind regards,

Zsofi Zombor
